# A Unified Approach to Interpreting and Boosting Adversarial Transferability

**Xin Wang**[a*]**, Jie Ren**[a*]**, Shuyun Lin**[a]**, Xiangming Zhu**[a]**, Yisen Wang**[b]**, Quanshi Zhang**[a†]
[a]Shanghai Jiao Tong University
[b]Key Lab. of Machine Perception (MoE), School of EECS, Peking University, Beijing, China

## Abstract

In this paper, we use the interaction inside adversarial perturbations to explain and boost the adversarial transferability. We discover and prove the negative correlation between the adversarial transferability and the interaction inside adversarial perturbations. The negative correlation is further verified through different DNNs with various inputs. Moreover, this negative correlation can be regarded as a unified perspective to understand current transferability-boosting methods. To this end, we prove that some classic methods of enhancing the transferability essentially decease interactions inside adversarial perturbations. Based on this, we propose to directly penalize interactions during the attacking process, which significantly improves the adversarial transferability. Our code is available online[1].

## 1 Introduction

Adversarial examples of deep neural networks (DNNs) have attracted increasing attention in recent years (Ma et al., 2018; Madry et al., 2018; Wang et al., 2019; Ilyas et al., 2019; Duan et al., 2020; Wu et al., 2020b; Ma et al., 2021). Goodfellow et al. (2014) found the transferability of adversarial perturbations, and used perturbations generated on a source DNN to attack other target DNNs. Although many methods have been proposed to enhance the transferability of adversarial perturbations (Dong et al., 2018; Wu et al., 2018; 2020a), the essence of the improvement of the transferability is still unclear.

This paper considers the interaction inside adversarial perturbations as a new perspective to interpret adversarial transferability. Interactions inside adversarial perturbations are defined using the Shapley interaction index proposed in game theory (Michel & Marc, 1999; Shapley, 1953). Given an input sample $x \in \mathbb{R}^n$, the adversarial attack aims to fool the DNN by adding an imperceptible perturbation $\delta \in \mathbb{R}^n$ on $x$. Each unit in the perturbation map is termed a *perturbation unit*. Let $\phi_i$ denote the importance of the $i$-th perturbation unit $\delta_i$ to attacking. $\phi_i$ is implemented as the Shapley value, which will be explained later. The interaction between perturbation units $\delta_i, \delta_j$ is defined as the change of the $i$-th unit's importance $\phi_i$ when the $j$-th unit is perturbed *w.r.t* the case when the $j$-th unit is not perturbed. If the perturbation $\delta_j$ on the $j$-th unit increases the importance $\phi_i$ of the $i$-th unit, then there is a positive interaction between $\delta_i$ and $\delta_j$. If the perturbation $\delta_j$ decreases the importance $\phi_i$, it indicates a negative interaction.

In this paper, we discover and partially prove a clear negative correlation between the transferability and the interaction between adversarial perturbation units, *i.e.* adversarial perturbations with lower transferability tend to exhibit larger interactions between perturbation units. We verify such a correlation based on both the theoretical proof and comparative studies. Furthermore, based on the correlation, we propose to penalize interactions during attacking to improve the transferability.

---

[*]Equal contribution
[†]Correspondence. This study is conducted under the supervision of Dr. Quanshi Zhang. zqs1022@sjtu.edu.cn. Quanshi Zhang is with the John Hopcroft Center and the MoE Key Lab of Artificial Intelligence, AI Institute, at the Shanghai Jiao Tong University, China.

[1]https://github.com/xherdan76/A-Unified-Approach-to-Interpreting-and-Boosting-Adversarial-Transferability

In fact, our research group led by Dr. Quanshi Zhang has proposed game-theoretic interactions, including interactions of different orders (Zhang et al., 2020) and multivariate interactions (Zhang et al., 2021c). As a basic metric, the interaction can be used to explain signal processing in trained DNNs from different perspectives. For example, we have build up a tree structure to explain the hierarchical interactions between words in NLP models (Zhang et al., 2021a). We have also used interactions to explain the generalization power of DNNs (Zhang et al., 2021b). The interaction can also explain the utility of adversarial training (Ren et al., 2021). As an extension of the system of game-theoretic interactions, in this study, we explain the adversarial transferability based on interactions.

In this paper, the background for us to investigate the correlation between adversarial transferability and the interaction is as follows. First, we prove that multi-step attacking usually generates perturbations with larger interactions than single-step attacking. Second, according to (Xie et al., 2019), multi-step attacking tends to generate more over-fitted adversarial perturbations with lower transferability than single-step attacking. We consider that the more dedicated interaction reflects more over-fitting towards the source DNN, which hurts adversarial transferability. In this way, we propose the hypothesis that *the transferability and the interaction are negatively correlated.*

• **Comparative studies** are conducted to verify this negative correlation through different DNNs.

• **Unified explanation.** Such a negative correlation provides a unified view to understand current transferability-boosting methods. We theoretically prove that some classic transferability-boosting methods (Dong et al., 2018; Wu et al., 2018; 2020a) essentially decrease interactions between perturbation units, which also verifies the hypothesis of the negative correlation.

• **Boosting adversarial transferability.** Based on above findings, we propose a loss to decrease interactions between perturbation units during attacking, namely the interaction loss, in order to enhance the adversarial transferability. The effectiveness of the interaction loss further proves the negative correlation between the adversarial transferability and the interaction inside adversarial perturbations. Furthermore, we also try to only use the interaction loss to generate perturbations without the loss for the classification task. We find that such perturbations still exhibit moderate adversarial transferability for attacking. Such perturbations may decrease interactions encoded by the DNN, thereby damaging the inference patterns of the input.

Our contributions are summarized as follows. (1) We reveal the negative correlation between the transferability and the interaction inside adversarial perturbations. (2) We provide a unified view to understand current transferability-boosting methods. (3) We propose a new loss to penalize interactions inside adversarial perturbations and enhance the adversarial transferability.

## 2 RELATED WORK

**Adversarial transferability.** Attacking methods can be roughly divided into two categories, *i.e.* white-box attacks (Szegedy et al., 2013; Goodfellow et al., 2014; Papernot et al., 2016; Carlini & Wagner, 2017; Kurakin et al., 2017; Su et al., 2017; Madry et al., 2018) and black-box attacks (Liu et al., 2016; Papernot et al., 2017; Chen et al., 2017a; Bhagoji et al., 2018; Ilyas et al., 2018; Bai et al., 2020). A specific type of the black-box attack is based on the adversarial transferability (Dong et al., 2018; Wu et al., 2018; Xie et al., 2019; Wu et al., 2020a), which transfers adversarial perturbations on a surrogate/source DNN to a target DNN.

Thus, some previous studies focused on the transferability of adversarial attacking. Liu et al. (2016) demonstrated that non-targeted attacks were easy to transfer, while the targeted attacks were difficult to transfer. Wu et al. (2018) and Demontis et al. (2019) explored factors influencing the transferability, such as network architectures, model capacity, and gradient alignment. Several methods have been proposed to enhance the transferability of adversarial perturbations. The momentum iterative attack (MI Attack) (Dong et al., 2018) incorporated the momentum of gradients to boost the transferability. The variance-reduced attack (VR Attack) (Wu et al., 2018) used the smoothed gradients to craft perturbations with high transferability. The diversity input attack (DI Attack) (Xie et al., 2019) applied the adversarial attacking to randomly transformed input images, which included random resizing and padding with a certain probability. The skip gradient method (SGM Attack) (Wu et al., 2020a) used the gradients of the skip connection to improve the transferability. Dong et al. (2019) proposed the translation-invariant attack (TI Attack) to evade robustly trained DNNs. Li et al.

(2020) used the dropout erosion and the skip connection erosion to improve the transferability. In comparison, we explain the transferability based on game theory, and discover the negative correlation between the transferability and interactions as a unified explanation for some above methods.

**Interaction.** The interaction between input variables has been widely investigated. Michel & Marc (1999) proposed the Shapley interaction index based on the Shapley value (Shapley, 1953) in game theory. Daria Sorokina (2008) defined the interaction of $K$ input variables of additive models. Scott Lundberg (2017) quantified interactions between each pair of input variables for tree-ensemble models. Some studies mainly focused on interactions to analyze DNNs. Tsang et al. (2018) measured statistical interactions based on DNN weights. Murdoch et al. (2018) proposed to extract interactions in LSTMs by disambiguating information of different gates, and Singh et al. (2019) extended this method to CNNs. Jin et al. (2020) quantified the contextual independence of words to hierarchically explain the LSTMs. Janizek et al. (2020) extended the method of Integrated Gradients (Sundararajan et al., 2017) to quantify pairwise interactions of input features based on the Hessian matrix, which required the DNN to use the SoftPlus operation replace the ReLU operation. Chen et al. (2020) extended the attribution in (Chen & Ji, 2020) to use the Shapley interaction index to generate hierarchical explanations of NLP tasks. In comparison, in this study, we use the Shapley interaction index to explain and improve the transferability of adversarial perturbations.

## 3 THE RELATIONSHIP BETWEEN TRANSFERABILITY AND INTERACTIONS

**Preliminaries: the Shapley value.** The Shapley value was first proposed in game theory (Shapley, 1953). Considering multiple players in a game, each player aims to win a high reward. The Shapley value is considered as a unique and unbiased approach to fairly allocating the total reward gained by all players to each player (Weber, 1988). The Shapley value satisfies four desirable properties, *i.e.* the *linearity*, *dummy*, *symmetry*, and *efficiency* (please see the Appendix A.1 for details). Let $\Omega = \{1, 2, \ldots, n\}$ denote the set of all players, and let $v(\cdot)$ denote the reward function. $v(S)$ represents the reward obtained by a set of players $S \subseteq \Omega$. The Shapley value $\phi(i|\Omega)$ unbiasedly measures the contribution of the $i$-th player to the total reward gained by all players in $\Omega$, as follows.

$$\sum_i \phi(i|\Omega) = v(\Omega) - v(\emptyset), \qquad \phi(i|\Omega) = \sum_{S \subseteq \Omega \setminus \{i\}} \frac{|S|!\,(n - |S| - 1)!}{n!} (v(S \cup \{i\}) - v(S)). \quad (1)$$

**Adversarial attack.** Given an input sample $x \in [0, 1]^n$ with the true label $y \in \{1, 2, \ldots, C\}$, we use $h(x) \in \mathbb{R}^C$ to denote the output of the DNN before the softmax layer. To simplify the story, in this study, we mainly focus on the untargeted adversarial attack. The goal of the untargeted adversarial attack is to add a human-imperceptible perturbation $\delta \in \mathbb{R}^n$ on the sample $x$, and make the DNN classify the perturbed sample $x' = x + \delta$ into an incorrect category, *i.e.* $\arg\max_{y'} h_{y'}(x') \neq y$. The objective of adversarial attacking is usually formulated as follows.

$$\underset{\delta}{\text{maximize}} \quad \ell(h(x + \delta), y) \quad \text{s.t.} \quad \|\delta\|_p \leq \epsilon, \ x + \delta \in [0, 1]^n, \tag{2}$$

where $\ell(h(x + \delta), y)$ is referred to the classification loss, and $\epsilon$ is a constant of the norm constraint. Please see Appendix C for technical details of solving Equation (2).

### 3.1 THEORETICAL UNDERSTANDING OF THE ADVERSARIAL ATTACK IN GAME THEORY.

In adversarial attacking, given the perturbation $\delta \in \mathbb{R}^n$, we use $\Omega = \{1, 2, \ldots, n\}$ to denote all units/dimensions in the perturbation. We use the Shapley value in Equation (1) to measure the contribution of each perturbation unit $i \in \Omega$ to the attack. To this end, it requires us to define the utility of a subset of perturbation units $S \subseteq \Omega$ for attacking, which can be formulated as $v(S) = \max_{y' \neq y} h_{y'}(x + \delta^{(S)}) - h_y(x + \delta^{(S)})$, according to Equation (2). $h_y(\cdot)$ is the value of the $y$-th element of $h(\cdot) \in \mathbb{R}^C$. $\delta^{(S)} \in \mathbb{R}^n$ is the perturbation which only contains perturbation units in $S$, *i.e.* $\forall i \in S, \delta_i^{(S)} = \delta_i$; $\forall i \notin S, \delta_i^{(S)} = 0$. In this way, $v(\Omega) = \max_{y' \neq y} h_{y'}(x + \delta) - h_y(x + \delta)$ denotes the utility of all perturbation units, and $v(\emptyset) = \max_{y' \neq y} h_{y'}(x) - h_y(x)$ denotes the baseline score without perturbations. Thus, the overall contribution of perturbation units can be measured as $v(\Omega) - v(\emptyset)$. We apply the Shapley value in Equation (1) to assign the overall contribution to each perturbation unit as $\sum_i \phi(i|\Omega) = v(\Omega) - v(\emptyset)$, where $\phi(i|\Omega)$ denotes the contribution of the $i$-th perturbation unit.

**Interactions.** Perturbation units do not contribute to the adversarial utility independently. For example, perturbation units may form a certain pattern, *e.g.* an edge in the image. Thus, perturbations units in the edge must appear together. The absence of a few units in the pattern may invalidate this pattern. Let us consider two perturbation units $i, j$. According to (Michel & Marc, 1999), the Shapley interaction index between units $i, j$ is defined as the additional contribution as follows.

$$I_{ij}(\delta) = \phi(S_{ij}|\Omega') - [\phi(i|\Omega \setminus \{j\}) + \phi(j|\Omega \setminus \{i\})], \tag{3}$$

where $\phi(i|\Omega \setminus \{j\})$ and $\phi(j|\Omega \setminus \{i\})$ represent the individual contributions of units $i$ and $j$, respectively, when the perturbation units $i, j$ work individually. Note that $\phi(i|\Omega \setminus \{j\})$ is computed in the scenario of considering the unit $j$ always absent. $\sum_i \phi(i|\Omega \setminus \{j\}) = v(\Omega \setminus \{j\}) - v(\emptyset)$, due to the absence of perturbation unit $j$. $\phi(S_{ij}|\Omega')$ denotes the joint contribution of $i, j$, when perturbation units $i, j$ are regarded as a singleton unit $S_{ij} = \{i, j\}$. In this case, units $i, j$ are supposed to be always perturbed or not perturbed simultaneously, and we can consider that there are only $n - 1$ players in the game. Thus, the set of all perturbation units is considered as $\Omega' = \Omega \setminus \{i, j\} \cup S_{ij}$. The joint contribution of $S_{ij}$ is denoted by $\phi(S_{ij}|\Omega')$, s.t. $\sum_{i' \in \Omega' \setminus \{S_{ij}\}} \phi(i'|\Omega') + \phi(S_{ij}|\Omega') = v(\Omega') - v(\emptyset)$.

The interaction defined in Equation (3) is equivalent to the change of the $i$-th unit's importance $\phi_i$ when the unit $j$ exists *w.r.t* the case when the unit $j$ is absent. Please see Appendix D for details.

If $I_{ij}(\delta) > 0$, it means $\delta_i$ and $\delta_j$ cooperate with each other, *i.e.* the interaction is positive; if $I_{ij}(\delta) < 0$, it means $\delta_i$ and $\delta_j$ conflict with each other, *i.e.* the interaction is negative. The absolute value of $|I_{ij}(\delta)|$ indicates the interaction strength. The interaction is symmetric that $I_{ij}(\delta) = I_{ji}(\delta)$.

We are given an input sample $x \in \mathbb{R}^n$ and a DNN $h(\cdot)$ trained for classification. With the definition of interactions, in adversarial attacking, we have the following propositions:

**Proposition 1.** *(Proof in Appendix E) The adversarial perturbation generated by the multi-step attack via gradient descent is given as $\delta_{multi}^m = \alpha \sum_{t=0}^{m-1} \nabla_x \ell(h(x + \delta_{multi}^t), y)$, where $\delta_{multi}^t$ denotes the perturbation after the $t$-th step of updating, and $m$ is referred to as the total number of steps. The adversarial perturbation generated by the single-step attack is given as $\delta_{single} = \alpha m \nabla_x \ell(h(x), y)$. Then, the expectation of interactions between perturbation units in $\delta_{multi}^m$, $\mathbb{E}_{a,b}[I_{ab}(\delta_{multi}^m)]$, is larger than $\mathbb{E}_{a,b}[I_{ab}(\delta_{single})]$, i.e. $\mathbb{E}_{a,b}[I_{ab}(\delta_{multi}^m)] \geq \mathbb{E}_{a,b}[I_{ab}(\delta_{single})]$.*

Note that when we compare interactions inside different perturbations, magnitudes of these perturbations should be similar. It is because the comparison of interactions between adversarial perturbations of different magnitudes is not fair. Therefore, we use the step size $\alpha m$ in the single-step attack to roughly (not accurately) balance the magnitude of perturbations. The fairness is further discussed in Appendix E.1.

Proposition 1 shows that, in general, adversarial perturbations generated by the multi-step attack tend to exhibit larger interactions than those generated by the single-step attack. In addition, Appendix E.4 shows that the multi-step attack usually generates perturbations with larger interactions than noisy perturbations of the same magnitude. Besides, Xie et al. (2019) demonstrated that the multi-step attack tends to over-fit the source DNN, which led to low transferability. Intuitively, large interactions mean a strong cooperative relationship between perturbation units, which indicates the significant over-fitting towards adversarial perturbations oriented to the source DNN. In this way, we propose the hypothesis that *the adversarial transferability and the interactions inside adversarial perturbations are negatively correlated.*

## 3.2 EMPIRICAL VERIFICATION OF THE NEGATIVE CORRELATION

To verify the negative correlation between the transferability and interactions, we conduct experiments to examine whether adversarial perturbations with low transferability tend to exhibit larger interactions than those perturbations with high transferability. Given a source DNN and an input sample $x$, we generate the adversarial example $x' = x + \delta$. Then, given a target DNN $h^{(t)}$, we measure the transfer utility of $\delta$ as *Transfer Utility* $= [\max_{y' \neq y} h_{y'}^{(t)}(x + \delta) - h_y^{(t)}(x + \delta)] - [\max_{y' \neq y} h_{y'}^{(t)}(x) - h_y^{(t)}(x)]$ as mentioned in Section 3.1. The interaction is given as *Interaction* $= \mathbb{E}_{i,j}[I_{ij}(\delta)]$, which is computed on the source DNN. Note that the computational cost of $I_{ij}(\delta)$ is NP-hard. However, we prove that we can simplify the computation of the average interaction over all pairs of units as

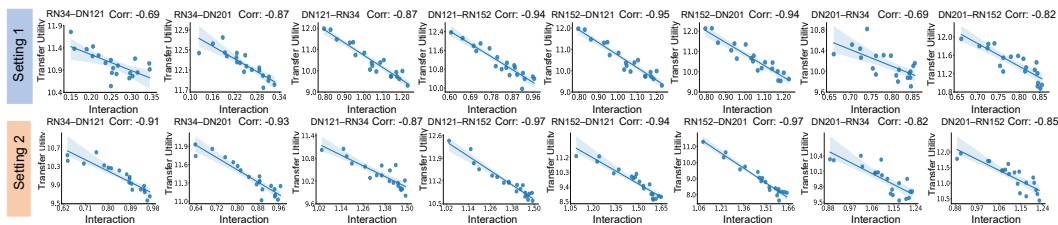

Figure 1: The negative correlation between the transfer utility and the interaction. The correlation is computed as the Pearson correlation. The blue shade in each subfigure represents the 95% confidence interval of the linear regression.

follows, which significantly reduces the computational cost. Please see Appendix F for the proof.

$$\mathbb{E}_{i,j}\left[I_{ij}(\delta)\right] = \frac{1}{n-1}\mathbb{E}_i\left[v(\Omega) - v(\Omega \setminus \{i\}) - v(\{i\}) + v(\emptyset)\right]. \tag{4}$$

Using 50 images randomly sampled from the validation set of the ImageNet dataset (Russakovsky et al., 2015), we generate adversarial perturbations on four types of DNNs, including ResNet-34/152(RN-34/152) (He et al., 2016) and DenseNet-121/201(DN-121/201) (Huang et al., 2017). We transfer adversarial perturbations generated on each ResNet to DenseNets. Similarly, we also transfer adversarial perturbations generated on each DenseNet to ResNets. Figure 1 shows the negative correlation between the transfer utility and the interaction. Each subfigure corresponds to a specific pair of source DNN and target DNN. In each subfigure, each point represents the average transfer utility and the average interaction of adversarial perturbations through all testing images. Different points represent the average interaction and the average transfer utility computed using different hyper-parameters. Given an input image $x$, adversarial perturbations are generated by solving the relaxed form of Equation (2) via the gradient descent, *i.e.* $\min_\delta -\ell(h(x+\delta), y) + c \cdot \|\delta\|_p^p$ s.t. $x + \delta \in [0,1]^n$, where $c \in \mathbb{R}$ is a scalar. In this way, we gradually change the value of $c$ and set different values of $p^2$ as different hyper-parameters to generate different adversarial perturbations, thereby drawing different points in each subfigure. Fair comparisons require adversarial perturbations generated with different hyper-parameters $c$ to be comparable with each other. Thus, we select a constant $\tau$ and take $\|\delta\|_2 = \tau$ as the stopping criteria of all adversarial attacks. Please see Appendix G for more details.

## 4 UNIFIED UNDERSTANDING OF TRANSFERABILITY-BOOSTING ATTACKS

In this section, we prove that some classical methods of improving the adversarial transferability essentially decrease interactions between perturbation units, although these methods are not originally designed to decrease the interaction. Without loss of generality, let us be given an input sample $x \in \mathbb{R}^n$ and a DNN $h(\cdot)$ trained for classification.

• **VR Attack** (Wu et al., 2018) smooths the classification loss with the Gaussian noise during attacking. In the VR Attack, the gradient of the input sample is computed as follows. $g^t = \mathbb{E}_{\xi \sim \mathcal{N}(0,\sigma^2 I)}\left[\nabla_x \ell(h(x + \delta^t + \xi), y)\right]$. The following proposition proves that the VR Attack is prone to decrease interactions inside perturbation units.

**Proposition 2.** *(Proof in Appendix H) The adversarial perturbation generated by the multi-step attack is given as $\delta_{multi}^m = \alpha \sum_{t=0}^{m-1} \nabla_x \ell(h(x + \delta_{multi}^t), y)$. The adversarial perturbation generated by the VR Attack is computed as $\delta_{vr}^m = \alpha \sum_{t=0}^{m-1} \nabla_x \hat{\ell}(h(x + \delta_{vr}^t), y)$, where $\hat{\ell}(h(x + \delta_{vr}^t), y) = \mathbb{E}_{\xi \sim \mathcal{N}(0,\sigma^2 I)}\left[\ell(h(x + \delta_{vr}^t + \xi), y)\right]$. Perturbation units of $\delta_{vr}^m$ tend to exhibit smaller interactions than $\delta_{multi}$, i.e. $\mathbb{E}_x \mathbb{E}_{a,b}[I_{ab}(\delta_{vr}^m)] \le \mathbb{E}_x \mathbb{E}_{a,b}[I_{ab}(\delta_{multi}^m)]$.*

Besides the theoretical proof, we also conduct experiments to compare interactions of perturbation units generated by the baseline multi-step attack (implemented as (Madry et al., 2018)) with those

---

[2]We set $p = 2$ as the setting 1, and $p = 5$ as the setting 2. To this end, the performance of adversarial perturbations is not the key issue in the experiment. Instead, we just randomly set the $p$ value to examine the trustworthiness of the negative correlation under various attacking conditions (even in extreme attacking conditions).

of perturbation units generated by the VR Attack. Table 5 shows that the VR Attack exhibits lower interactions between perturbation units than the baseline multi-step attack.

• **MI Attack** (Dong et al., 2018) incorporates the momentum of gradients when updating the adversarial perturbation. In the MI Attack, the gradient used in step $t$ is computed as follows. $g^t = \mu \cdot g^{t-1} + \nabla_x \ell \left( h \left( x + \delta^{t-1} \right), y \right) / \left\| \nabla_x \ell \left( h \left( x + \delta^{t-1} \right), y \right) \right\|_1$.

Note that the original MI Attack and the multi-step attack cannot be directly compared, since that magnitudes of the generated perturbations cannot be fairly controlled. The values of interactions are sensitive to the magnitude of perturbations. Comparing perturbations with different magnitudes is not fair. Thus, we slightly revise the MI Attack as $\forall t > 0, g_{mi}^t = \mu g_{mi}^{t-1} + (1 - \mu) \nabla_x \ell (h(x + \delta_{mi}^{t-1}), y); g_{mi}^0 = 0$, where $\mu = (t - 1)/t$. We investigate the interaction of adversarial perturbations generated by the original multi-step attack and the MI Attack. We prove the following proposition, which shows that the MI Attack decreases the interaction between perturbation units in most cases.

**Proposition 3.** *(Proof in Appendix I) The adversarial perturbation generated by the multi-step attack is given as $\delta_{multi}^m = \alpha \sum_{t=0}^{m-1} \nabla_x \ell(h(x + \delta_{multi}^t), y)$. The adversarial perturbation generated by the multi-step attack incorporating the momentum is computed as $\delta_{mi}^m = \alpha \sum_{t=0}^{m-1} g_{mi}^t$. Perturbation units of $\delta_{mi}^m$ exhibit smaller interactions than $\delta_{multi}^m$, i.e. $\mathbb{E}_{a,b}[I_{ab}(\delta_{mi}^m)] \leq \mathbb{E}_{a,b}[I_{ab}(\delta_{multi}^m)]$.*

• **SGM Attack** (Wu et al., 2020a) exploits the gradient information of the skip connection in ResNets to improve the transferability of adversarial perturbations. The SGM Attack revises the gradient in the backpropagation, which can be considered as to add a specific dropout operation in the backpropagation. We notice that Zhang et al. (2021b) has proved that the dropout operation can decrease the significance of interactions, so as to decrease the significance of the over-fitting of DNNs. Thus, this also proves that the SGM Attack decreases interactions between perturbation units.

Besides the theoretical proof, we also conduct experiments to compare interactions of perturbation units generated by the baseline multi-step attack (implemented as Madry et al. (2018)) with those of perturbation units generated by the SGM Attack. Table 5 shows that the SGM Attack exhibits lower interactions than the baseline multi-step attack.

## 5 THE INTERACTION LOSS FOR TRANSFERABILITY ENHANCEMENT

**Interaction loss.** Based on findings in previous sections, we propose a loss to directly penalize interactions during attacking, in order to improve the transferability of adversarial perturbations. Based on Equation (2), we jointly optimize the classification loss and the interaction loss to generate adversarial perturbations. This method is termed the interaction-reduced attack (IR Attack).

$$\max_\delta \left[ \ell(h(x + \delta), y) - \lambda \ell_{\text{interaction}} \right], \quad \ell_{\text{interaction}} = \mathbb{E}_{i,j} \left[ I_{ij}(\delta) \right] \quad \text{s.t. } \|\delta\|_p \leq \epsilon, \ x + \delta \in [0, 1]^n, \quad (5)$$

where $\ell_{\text{interaction}}$ is the interaction loss, and $\lambda$ is a constant weight for the interaction loss. Although the computation of the interaction loss can be simplified according to Equation (4), the computational cost of the interaction loss is intolerable, when the dimension of images is high. Therefore, as a trade-off between the accuracy and the computational cost, we divide the input image into $16 \times 16$ grids. We measure and penalize interactions at the grid level, instead of the pixel level. Moreover, we apply an efficient sampling method to approximate the expectation operation during the computation of interactions in Equation (4). Figure 2 visualizes interactions between adjacent perturbation units at the grid level generated with and without the interaction loss.

**Experiments.** For implementation, we generated adversarial perturbations on six different source DNNs, including Alexnet (Krizhevsky et al., 2012), VGG-16 (Simonyan & Zisserman, 2015), ResNet-34/152 (RN-34/152) (He et al., 2016) and DenseNet-121/201 (DN-121/201) (Huang et al., 2017). For each source DNN, we tested the transferability of the generated perturbations on seven target DNNs, including VGG-16, ResNet-152 (RN-152), DenseNet-201 (DN-201), SENet-154 (SE-154) (Hu et al., 2018), InceptionV3 (IncV3) (Szegedy et al., 2016), InceptionV4 (IncV4) (Szegedy et al., 2017), and Inception-ResNetV2 (IncResV2) (Szegedy et al., 2017). In addition, three state-of-the-art DNNs, including the Dual-Path-Network (DPN-68) (Chen et al., 2017b), the NASNet-LARGE (NASN-L) (Zoph et al., 2018), and the Progressive NASNet (PNASN) (Liu et al., 2018), were used as target DNNs to evaluate the ensemble source model (will be introduced in the next paragraph). Besides unsecured target DNNs mentioned above, we also used three secured target

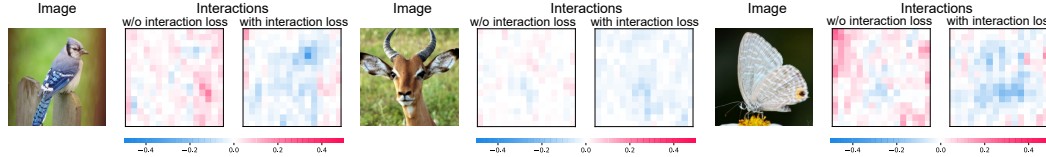

Figure 2: Visualization of interactions between neighboring perturbation units generated with and without the interaction loss. The color in the visualization is computed as $color[i] \propto E_{j \in N_i}\left[I_{ij}(\delta)\right]$, where $N_i$ denotes the set of adjacent perturbation units of the perturbation unit $i$. Here, we ignore interactions between non-adjacent units to simplify the visualization. It is because adjacent units usually encode much more significant interactions than other units. The interaction loss forces the perturbation to encode more negative interactions.

Table 1: The success rates of $L_\infty$ and $L_2$ black-box attacks crafted on six source models, including AlexNet, VGG16, RN-34/152, DN-121/201, against seven target models. Transferability of adversarial perturbations can be enhanced by penalizing interactions.

| Source | Method | VGG-16 | RN152 | DN-201 | SE-154 | IncV3 | IncV4 | IncResV2 |
|---|---|---|---|---|---|---|---|---|
| AlexNet | PGD $L_\infty$ | 67.0±1.6 | 27.8±1.1 | 32.3±0.4 | 28.2±0.7 | 29.1±1.5 | 23.0±0.4 | 18.6±1.5 |
| | PGD $L_\infty$+IR | **78.7±1.0** | **42.0±1.5** | **50.3±0.4** | **41.2±0.6** | **43.7±0.5** | **36.4±1.5** | **29.0±1.0** |
| VGG-16 | PGD $L_\infty$ | – | 43.0±1.8 | 48.3±2.0 | 52.9±2.7 | 39.3±0.7 | 49.3±1.1 | 29.7±2.0 |
| | PGD $L_\infty$+IR | – | **63.1±1.6** | **70.0±1.1** | **71.2±1.5** | **57.6±1.0** | **68.6±3.2** | **49.2±1.2** |
| RN-34 | PGD $L_\infty$ | 65.4±2.9 | 59.2±2.7 | 63.5±3.3 | 33.1±2.9 | 27.4±3.6 | 23.9±1.7 | 21.1±1.1 |
| | PGD $L_\infty$+IR | **84.0±0.5** | **84.7±2.3** | **88.5±0.9** | **64.4±1.6** | **56.9±3.1** | **59.3±4.3** | **49.2±1.1** |
| RN-152 | PGD $L_\infty$ | 51.6±3.2 | – | 61.5±2.4 | 33.9±1.5 | 28.1±0.9 | 25.0±1.2 | 22.4±1.0 |
| | PGD $L_\infty$+IR | **72.3±1.2** | – | **82.1±1.3** | **61.1±0.9** | **53.6±0.8** | **50.6±3.5** | **46.0±2.3** |
| DN-121 | PGD $L_\infty$ | 68.6±1.1 | 63.6±3.2 | 86.9±1.5 | 46.1±1.5 | 37.3±1.6 | 37.1±2.1 | 28.9±2.8 |
| | PGD $L_\infty$+IR | **85.0±0.3** | **84.8±0.4** | **95.1±0.2** | **70.3±1.7** | **61.1±2.5** | **62.1±2.0** | **53.5±0.3** |
| DN-201 | PGD $L_\infty$ | 64.4±1.4 | 67.8±0.2 | – | 50.9±0.8 | 39.5±3.3 | 36.5±0.9 | 34.2±0.4 |
| | PGD $L_\infty$+IR | **78.6±2.5** | **85.0±1.1** | – | **73.9±0.5** | **61.6±1.8** | **63.7±0.6** | **56.4±2.1** |
| AlexNet | PGD $L_2$ | 85.1±1.5 | 58.9±1.0 | 60.2±2.1 | 55.1±1.5 | 56.0±3.7 | 49.6±3.4 | 44.6±3.3 |
| | PGD $L_2$+IR | **91.6±1.1** | **72.0±1.6** | **76.8±1.0** | **69.0±1.0** | **73.0±0.8** | **63.1±2.1** | **59.4±1.9** |
| VGG-16 | PGD $L_2$ | – | 76.7±0.9 | 82.3±2.9 | 83.5±1.9 | 77.5±3.6 | 82.1±2.2 | 69.4±2.1 |
| | PGD $L_2$+IR | – | **86.5±0.9** | **88.9±1.5** | **89.6±1.2** | **85.2±1.1** | **88.3±1.4** | **80.4±0.4** |
| RN-34 | PGD $L_2$ | 88.2±1.4 | 86.2±0.4 | 89.6±1.3 | 66.9±1.1 | 64.2±2.9 | 60.0±1.9 | 55.2±1.8 |
| | PGD $L_2$+IR | **95.2±0.2** | **95.4±0.1** | **96.7±0.6** | **86.7±1.2** | **84.3±0.6** | **81.8±1.9** | **80.4±1.9** |
| DN-121 | PGD $L_2$ | 89.4±1.1 | 86.8±1.0 | 97.6±1.0 | 75.6±1.7 | 70.1±2.9 | 70.4±4.4 | 66.5±4.7 |
| | PGD $L_2$+IR | **94.2±0.1** | **93.3±0.8** | **97.7±0.3** | **87.8±0.7** | **84.5±0.7** | **84.2±0.1** | **82.4±0.1** |

Table 2: The success rates of $L_\infty$ black-box attacks crafted on the ensemble model (RN-34+RN-152+DN-121) against nine target models.

| Source | Method | VGG-16 | RN-152 | DN-201 | SE-154 | IncV3 | IncV4 | IncResV2 | DPN-68 | NASN-L | PNASN |
|---|---|---|---|---|---|---|---|---|---|---|---|
| Ensemble | PGD $L_\infty$ | 86.6±1.2 | **99.9±0.1** | **97.0±0.7** | 70.7±1.6 | 64.2±0.3 | 57.7±2.4 | 53.1±0.7 | 61.6±0.5 | 59.6±0.4 | 72.3±0.3 |
| | PGD $L_\infty$+IR | **91.5±0.1** | 92.4±1.6 | 92.1±1.7 | **86.1±0.3** | **81.6±0.9** | **79.9±1.7** | **78.4±1.3** | **82.5±1.0** | **82.3±1.6** | **85.6±0.5** |

models for testing, which were learned via ensemble adversarial training: IncV3$_{ens3}$ (ensemble of three IncV3 networks), IncV3$_{ens4}$ (ensemble of four IncV3 networks), and IncResV2$_{ens3}$ (ensemble of three IncResV2 networks), which were released by Tramèr et al. (2017).

*Ensemble source model:* Besides above adversarial transferring from a single-source model, we also conducted the proposed IR Attack in the scenario of the ensemble-based attacking (Liu et al., 2016), in order to generate adversarial perturbations on the ensemble of RN-34, RN-152, and DN-121.

*Baselines.* The first baseline method, the PGD Attack (Madry et al., 2018), directly solved the Equation (2), which was widely used for adversarial attacks. Besides this baseline attack, the other four baselines were the MI Attack (Dong et al., 2018), the VR Attack (Wu et al., 2018), the SGM Attack (Wu et al., 2020a), and the TI Attack (Dong et al., 2019). Our method was implemented according to Equation (5), namely the IR Attack. Because the SGM Attack was one of the top-ranked methods of boosting the adversarial transferability, we further added the interaction loss $\ell_{\text{interaction}}$ to the SGM Attack as another implementation of our method (namely the SGM+IR Attack). We also used the interaction loss to boost the performance of the MI Attack and the VR Attack (namely MI+IR and VR+IR, respectively). Please see Appendix M.1 for details. Moreover, as Section 4 states, the MI Attack, VR Attack, and SGM Attack also decrease interactions during attacking. Thus, we combined the IR Attack with all these interaction-reducing techniques together as a new implementation of our method, namely the HybridIR Attack. All attacks were conducted with 100

Table 3: Transferability against the secured models: the success rates of $L_\infty$ black-box attacks crafted on RN-34 and DN-121 source models against three secured models.

| Source | Method | IncV3$_{ens3}$ | IncV3$_{ens4}$ | IncRes$_{ens3}$ | Source | Method | IncV3$_{ens3}$ | IncV3$_{ens4}$ | IncRes$_{ens3}$ |
|--------|--------|------|------|------|--------|--------|------|------|------|
| RN-34 | PGD $L_\infty$ | 9.8±0.1 | 10.0±0.5 | 5.7±0.3 | DN-121 | PGD $L_\infty$ | 12.8±0.1 | 11.2±1.7 | 6.9±1.0 |
| | PGD $L_\infty$+IR | **26.5±2.9** | **22.1±1.3** | **14.3±0.4** | | PGD $L_\infty$+IR | **28.0±1.8** | **26.5±2.1** | **17.4±1.3** |
| | TI[4] | 21.4±0.8 | 20.9±0.9 | 14.9±1.4 | | TI[4] | 26.8±1.3 | 26.1±1.5 | 19.4±1.6 |
| | TI[4] + IR | **33.6±0.4** | **33.2±0.3** | **24.0±0.5** | | TI[4] + IR | **38.0±2.5** | **42.2±7.7** | **29.0±1.4** |

Table 4: The success rates of $L_\infty$ black-box attacks crafted by different methods on four source models (RN-34/152, DN-121/201) against seven target models. Transferability of adversarial perturbations can be enhanced by penalizing interactions.

| Source | Method | VGG-16 | RN152 | DN-201 | SE-154 | IncV3 | IncV4 | IncResV2 |
|--------|--------|--------|-------|--------|--------|-------|-------|----------|
| RN-34 | MI | 80.1±0.5 | 73.0±2.3 | 77.7±0.5 | 48.9±0.8 | 46.2±1.2 | 39.9±0.5 | 34.8±2.5 |
| | VR | 88.8±0.2 | 86.4±1.6 | 87.9±2.4 | 62.1±1.5 | 58.4±3.0 | 56.3±2.3 | 49.7±0.9 |
| | SGM | 91.8±0.6 | 89.0±0.9 | 90.0±0.4 | 68.0±1.4 | 63.9±0.3 | 58.2±1.1 | 54.6±1.2 |
| | SGM+IR | 94.7±0.6 | 91.7±0.6 | 93.4±0.8 | 72.7±0.4 | 68.9±0.9 | 64.1±1.3 | 61.3±1.0 |
| | HybridIR | **96.5±0.1** | **94.9±0.3** | **95.6±0.6** | **79.7±1.0** | **77.1±0.8** | **73.8±0.1** | **70.2±0.5** |
| RN-152 | MI | 70.3±0.6 | – | 74.8±1.4 | 51.7±0.8 | 47.1±0.9 | 40.5±1.6 | 36.8±2.7 |
| | VR | 83.9±3.4 | – | 91.1±0.9 | 70.0±3.7 | 63.1±0.9 | 58.8±0.1 | 56.2±1.3 |
| | SGM | 88.2±0.5 | – | 90.2±0.3 | 72.7±1.4 | 63.2±0.7 | 59.1±1.5 | 58.1±1.2 |
| | SGM+IR | 92.0±1.0 | – | 92.5±0.4 | 79.3±0.1 | 69.6±0.8 | 66.2±1.0 | 63.6±0.9 |
| | HybridIR | **95.3±0.4** | – | **96.9±0.2** | **84.7±0.7** | **80.0±1.2** | **77.5±0.8** | **75.6±0.6** |
| DN-121 | MI | 83.0±4.9 | 72.0±0.7 | 91.5±0.2 | 58.4±2.6 | 54.6±1.6 | 49.2±2.4 | 43.9±1.5 |
| | VR | 91.5±0.5 | 88.7±0.5 | 98.8±0.2 | 75.1±1.3 | 74.3±1.7 | 75.6±3.0 | 69.8±1.3 |
| | SGM | 88.7±0.9 | 88.1±1.0 | 98.0±0.4 | 78.0±0.9 | 64.7±2.5 | 65.4±2.3 | 59.7±1.7 |
| | SGM+IR | 91.7±0.2 | 90.4±0.4 | 94.3±0.1 | 87.0±0.4 | 78.8±1.3 | 79.5±0.2 | 75.8±2.7 |
| | HybridIR | **96.9±0.4** | **96.8±0.4** | **99.1±0.4** | **90.9±0.5** | **88.4±0.8** | **87.8±0.8** | **87.1±0.4** |
| DN-201 | MI | 77.3±0.8 | 74.8±1.4 | – | 64.6±1.0 | 56.5±2.5 | 51.1±2.1 | 47.8±1.9 |
| | VR | 87.3±1.1 | 90.4±1.2 | – | 78.0±1.5 | 75.8±2.1 | 75.8±1.3 | 71.3±1.2 |
| | SGM | 87.3±0.3 | 92.4±1.0 | – | 82.9±0.2 | 72.3±0.3 | 71.3±0.6 | 68.8±0.5 |
| | SGM+IR | 89.5±0.9 | 91.8±0.7 | – | 87.3±1.2 | 82.5±0.8 | 80.3±0.3 | 81.5±0.5 |
| | HybridIR | **94.4±0.1** | **96.9±0.5** | – | **91.7±0.2** | **89.6±0.6** | **88.3±0.3** | **87.3±0.7** |

steps[3] on randomly selected 1000 images of the validation set in the ImageNet dataset. We set $\epsilon = 16/255$ for the $L_\infty$ attack, and set $\epsilon = 16/255\sqrt{n}$ following the setting in (Dong et al., 2018) for the $L_2$ attack. The step size was set to $2/255$ for all attacks. Considering the efficiency of signal processing in DNNs with different depths, we set $\lambda = 1$ for the IR Attack, when the source DNN was ResNet. We set $\lambda = 2$, for other source DNNs. To enable fair comparisons, the transferability of each baseline was computed based on the best adversarial perturbation during the 100 steps via the leave-one-out (LOO) validation. Please see the Appendix K for the motivation and the evidence of the LOO evaluation of transferability. All attacks were conducted with three different random samplings of grids or different initial perturbations.

Table 1 reports the success rates of the baseline attack (PGD (Madry et al., 2018)) and the IR Attack, namely PGD $L_\infty$+IR of $L_\infty$ attacks and PGD $L_2$+IR of $L_2$ attacks. Compared with the baseline attack, the transferability was significantly improved by the interaction loss on various source models against different target models. Let us focus on the $L_\infty$ attack. For most source models and target models, the transferability enhancement brought by the interaction loss was more than 10%. In particular, when the source DNN and the target DNN were DN-201 and IncV4, respectively, the baseline attack achieved the transferability of 36.5%. With the interaction loss, the transferability was improved to 63.7% (> 27% gain). As Table 2 shows, in most cases, the IR Attack on the ensemble model generated more transferable perturbations than the PGD Attack. Besides, as Table 3 shows, our interaction loss also improved the transferability against the secured target DNNs. Such improvement further verified the negative correlation between transferability and interactions. Note that we did not use the LOO in Table 3, in order to make experimental settings in this table consistent with the evaluation used by Tramèr et al. (2017). Table 4 shows the improvement of the transferability obtained by the interaction loss on other attacking methods. The interaction loss could further boost the transferability of state-of-the-art transfer attacks. Without the interaction loss, the highest transferability made by the SGM Attack against the IncResV2 was 68.8% (when the source is DN-201). When the interaction loss was added, the transferability was improved to 81.5% (>

---

[3]Previous studies usually set the number of steps to 10 or 20. Here, we set the number of steps to 100 together with the leave-one-out validation for fair comparisons of different attacks.

[4]The TI Attack was designed oriented to the secured DNNs which were robustly trained via adversarial training. Thus, we applied the TI Attack to the secured models in Table 3.

Figure 3: (a) The success rates of black-box attacks with the IR Attack using different values of $\lambda$. The success rates increased, when the value of $\lambda$ increased. (b) The transferability of adversarial perturbations generated by only using the interaction loss (without the classification loss). Such adversarial perturbations still exhibited moderate adversarial transferability. Points localized at the last epoch represent the transferability of noise perturbations as the baseline.

12% gain). Moreover, the HybridIR Attack, which combined all methods of reducing interactions together, improved success rates from the range of 54.6%~98.8% to the range of 70.2%~99.1%.

We can understand behaviors of the proposed interaction loss as follows. Different methods generate adversarial perturbations in different manifolds, thereby exhibiting different transferability. Based on the current perturbation, the interaction loss can point out the optimization direction towards further decrease of interactions in a local manner due to its optimization power. Thus, the interaction loss further boosts the transferability.

To further demonstrate the broad applicability of the interaction loss, besides untargeted attacks on the ImageNet dataset, we also conducted targeted attacks on the CIFAR-10 dataset (Krizhevsky & Hinton, 2009). Experimental results consistently showed that the adversarial transferability can be enhanced by reducing interactions in targeted attacks. Please see Appendix. M.2 for details.

*Effects of the interaction loss.* We tested the transferability of perturbations generated by the IR Attack with different weights of the interaction loss $\lambda$. In particular, the baseline attack (PGD) can be considered as the IR Attack when $\lambda = 0$. We conducted attacks on two source DNNs (RN-34, DN-121), and transferred adversarial perturbations to seven target DNNs (VGG16, RN-152, DN-201, SE-154, IncV3, IncV4, IncResV2). The attacks were conducted with 100 steps[3] on validation images in ImageNet . Figure 3 (a) shows the black-box success rates with different values of $\lambda$. The transferability of the IR Attack increased along with the increase of the weight $\lambda$.

*Attack only with the interaction loss.* To further understand the effects of the interaction loss, we generated perturbations by exclusively using the interaction loss (without the classification loss). We used the RN-34 and DN-121 as source DNNs and tested the transferability on seven target DNNs. The attacks were conducted with 100 steps[3] on ImageNet validation images. Figure 3 (b) shows the curve of the transferability in different epochs. We compared such adversarial perturbations with noise perturbations generated as $\epsilon \cdot \text{sign}(noise)$, where $noise \sim \mathcal{N}(0, \sigma^2 I)$, and $\epsilon = 16/255$, which was the same as the value used in the $L_\infty$ attack. We found that perturbations generated by only using the interaction loss still exhibited moderate adversarial transferability. This phenomenon may be explained as that such perturbations decrease most interactions in the DNN, thereby damaging the inference patterns in the input image.

## 6 CONCLUSION

In this paper, we have analyzed the transferability of adversarial perturbations from the perspective of interactions based on game theory. We have proved that the multi-step attack tends to generate adversarial perturbations with large interactions. We have discovered and partially proved the negative correlation between the transferability and interactions inside adversarial perturbations. *I.e.* adversarial perturbations with higher transferability usually exhibit more negative interactions. We have proved that some classical methods of enhancing the transferability essentially decrease interactions between perturbation units, which provides a unified view to understand the enhancement of transferability. Moreover, we have proposed a new loss to directly penalize interactions between perturbation units during attacking, which significantly improves the transferability of previous methods. Furthermore, we have found that adversarial perturbations generated only using the interaction loss without the classification loss still exhibited moderate transferability, which provides a new perspective to understand the transferability of adversarial perturbations.

## ACKNOWLEDGMENTS

All members in Shanghai Jiao Tong university, including Xin Wang, Jie Ren, Shuyun Lin, Xiangming Zhu, and Dr. Quanshi Zhang are supported by National Natural Science Foundation of China (61906120 and U19B2043) and Huawei Technologies. Dr. Yisen Wang is partially supported by the National Natural Science Foundation of China under Grant 62006153, and CCF-Baidu Open Fund (OF2020002). Xin Wang is supported by Wu Wen Jun Honorary Doctoral Scholarship, AI Institute, Shanghai Jiao Tong University.

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

## A    MOTIVATIONS FOR USING THE SHAPLEY INTERACTION INDEX

In this section, we discuss the motivations of using the Shapley interaction index to define the inter-action.

### A.1    FOUR PROPERTIES OF SHAPLEY VALUES

Let $\Omega = \{1, 2, \ldots, n\}$ denote the set of all players, and the reward function is $v$. Without ambiguity, we use $\phi(i|\Omega)$ to denote the Shapley value of the player $i$ in the game with all players $\Omega$ and reward function $v$, which is given as follows.

$$\phi(i|\Omega) = \sum_{S \subseteq \Omega \setminus \{i\}} \frac{|S|! \, (n - |S| - 1)!}{n!} (v(S \cup \{i\}) - v(S)). \tag{6}$$

The Shapley value satisfies the following four properties (Weber, 1988):

• *Linearity property:* If there are two games and the corresponding reward functions are $v$ and $w$, *i.e.* $v(S)$ and $w(S)$ measure the reward obtained by players in $S$ in these two games. Let $\phi_v(i|\Omega)$ and $\phi_w(i|\Omega)$ denote the Shapley value of the player $i$ in the game $v$ and game $w$, respectively. If these two games are combined into a new game, and the reward function becomes $reward(S) = v(S) + w(S)$, then the Shapley value comes to be $\phi_{v+w}(i|\Omega) = \phi_v(i|\Omega) + \phi_w(i|\Omega)$ for each player $i$ in $\Omega$.

• *Dummy property:* A player $i \in \Omega$ is referred to as a dummy player if $\forall S \subseteq \Omega \setminus \{i\}$, $v(S \cup \{i\}) = v(S) + v(\{i\})$. In this way, $\phi(i|\Omega) = v(\{i\}) - v(\emptyset)$, which means that player $i$ plays the game independently.

• *Symmetry property:* If $\forall S \subseteq \Omega \setminus \{i, j\}$, $v(S \cup \{i\}) = v(S \cup \{j\})$, then Shapley values of player $i$ and $j$ are equal, *i.e.* $\phi(i|\Omega) = \phi(j|\Omega)$ .

• *Efficiency property:* The sum of each individual's Shapley value is equal to the reward won by the coalition $N$, *i.e.* $\sum_i \phi(i|\Omega) = v(\Omega) - v(\emptyset)$. This property guarantees the overall reward can be allocated to each player in the game.

### A.2    MOTIVATIONS

**Theoretical rigor**. We use the Shapley interaction index defined based on the Shapley value, be-cause the Shapley value has a solid theoretical foundation in the game theory, which is the *unique* attribution satisfying the above four desirable axioms.

**Whether the metric depends on network architectures.** Because adversarial transferability is a general property for the attack, a convincing metric for adversarial transferability is supposed not to be directly related to the network architecture. To this end, the computation of the interaction defined on the Shapley value does not depend on the network architecture. In comparison, previous definitions of the interaction are usually oriented to model architectures. For example, the interaction proposed by Tsang et al. (2018) requires the DNN to be fully-connected. The two interaction metrics proposed by Murdoch et al. (2018) and Jin et al. (2020) are designed for LSTMs. The Hessian-based interaction (Janizek et al., 2020) requires the DNN to use the softPlus operation to replace the ReLU operation.

**Computational cost.** The computational cost of the Shapley-based interaction-reduction loss is relatively low. Because of the efficiency axiom of the Shapley value, we prove that the time cost of computing the interaction loss $\ell_{\text{interaction}} = \frac{1}{n-1} \mathbb{E}_i[v(\Omega) - v(\Omega \setminus \{i\}) - v(\{i\}) + v(\emptyset)]$ is linear, *i.e.* $O(n)$, where $n$ is the dimension of features. The linear complexity makes it possible to apply the interaction to high-dimensional data and deep neural networks. In contrast, the complexity of computing all possible pairwise interactions defined in (Daria Sorokina, 2008) is $O(n^2)$.

## B    COMPARISONS BETWEEN INTERACTIONS INSIDE PERTURBATIONS OF DIFFERENT ATTACKS

We have theoretically proved that some classical attacking methods of boosting the adversarial trans-ferability essentially decrease interactions inside perturbations. Besides the theoretical proof in Ap-

Table 5: The average interaction inside adversarial perturbations generated by different attacks.

| Method | RN-34 | RN-152 | DN-121 | DN-201 |
|---|---|---|---|---|
| Baseline (PGD Attack) | **0.422** | **0.926** | **0.909** | **0.784** |
| SGM Attack | -0.012 | 0.037 | 0.395 | 0.308 |
| VR Attack | 0.097 | 0.270 | 0.242 | 0.137 |

pendix I and Appendix H, we also conduct experiments to compare interactions of perturbation units when we generate adversarial perturbations with and without these attacking methods. Such experiments further verify that these methods of boosting the transferability essentially decrease interactions. We conduct attacks with the validation set in the ImageNet dataset on four DNNs, and measure the average interaction inside perturbation units. As Table 5 shows, the SGM Attack and the VR Attack decrease interactions inside perturbations.

## C  ADVERSARIAL ATTACK

In general, the objective of adversarial attacking can be formulated as the following optimization problem.

$$\underset{\delta}{\text{maximize}} \quad \ell(h(x + \delta), y) \quad \text{s.t.} \quad \|\delta\|_p \leq \epsilon, \ x + \delta \in [0, 1]^n, \tag{7}$$

where $\ell(h(x + \delta), y)$ is the classification loss. There are many ways to solve the above optimization problem under different norm constraints $\|\cdot\|_p$ (Goodfellow et al., 2014; Carlini & Wagner, 2017; Kurakin et al., 2017; Madry et al., 2018; Chen et al., 2018b; Wang et al., 2019).

**Optimization-based approach.** One approach to approximately solving Equation (7) is to solve the following relaxed form:

$$\underset{\delta}{\text{minimize}} \quad \{-\ell(h(x + \delta), y) + c \cdot \|\delta\|_p\} \quad \text{s.t.} \quad x + \delta \in [0, 1]^n, \tag{8}$$

where $c > 0$ is a scalar constant to balance the classification loss and the norm constraint. Szegedy et al. (2013); Carlini & Wagner (2017) have demonstrated the effectiveness of this method.

**Projected gradient descent (PGD) (Madry et al., 2018).** The PGD Attack is usually considered as one of the simplest and the most widely used baseline for adversarial attacking. In this paper, this method is called the *Baseline*. The PGD Attack directly optimizes the classification loss in Equation (7). Considering the norm constraint, after each step of updating, the PGD Attack projects the adversarial perturbation $\delta$ back to the $\epsilon$-ball, if the perturbation goes beyond the ball.

PGD updates adversarial perturbations in each step with the following equation:

$$\delta^{t+1} = \begin{cases} \Pi_\epsilon^{(\infty)} \left( \delta^t + \alpha \cdot \text{sign} \left( \nabla \ell \left( h \left( x + \delta^t \right), y \right) \right) \right), & p = +\infty \\ \Pi_\epsilon^{(2)} \left( \delta^t + \alpha \cdot \frac{\nabla \ell \left( h \left( x + \delta^t \right), y \right)}{\| \nabla \ell (h(x + \delta^t), y) \|_2} \right), & p = 2, \end{cases} \tag{9}$$

where $\delta^t$ denotes the perturbation of the $t$-th step. $\Pi_\epsilon^{(\infty)}$ and $\Pi_\epsilon^{(2)}$ are projection operations, which project the perturbation $\delta$ back to the $\epsilon$-ball, if the perturbation goes beyond the ball. $\alpha$ is the step size. Given $\delta \in \mathbb{R}^n$, we have:

$$\Pi_\epsilon^{(\infty)}(\delta_i) = \begin{cases} \epsilon \cdot \text{sign}(\delta_i), & \text{if } |\delta_i| > \epsilon \\ \delta_i, & \text{if } |\delta_i| \leq \epsilon \end{cases}, \qquad \Pi_\epsilon^{(2)}(\delta) = \begin{cases} \epsilon \frac{\delta}{\|\delta\|_2}, & \text{if } \|\delta\|_2 > \epsilon \\ \delta, & \text{if } \|\delta\|_2 \leq \epsilon \end{cases}. \tag{10}$$

## D  EQUIVALENT FORMS OF THE INTERACTION

In Section 3.1, the interaction between units $i, j$ is defined as the additional contribution as follows.

$$I_{ij}(\delta) = \phi(S_{ij}|\Omega') - [\phi(i|\Omega \setminus \{j\}) + \phi(j|\Omega \setminus \{i\})], \tag{11}$$

where $\phi(S_{ij}|\Omega')$ denotes the joint contribution of $i, j$, when perturbation units $i, j$ are regarded as a singleton unit $S_{ij} = \{i, j\}$, as follows.

$$\phi(S_{ij}|\Omega') = \sum_{S \subseteq \Omega \setminus \{i,j\}} \frac{|S|! \, (n - |S| - 2)!}{(n - 1)!} (v(S \cup \{i, j\}) - v(S)),$$

where $S_{ij} = \{i, j\}$ represents the coalition of perturbation units $i, j$. In this game, because perturbation units $i, j$ are regarded as a singleton player, we can consider there are only $n - 1$ players in the game, and consequently the set of players changes to $\Omega' = \Omega \setminus \{i, j\} \cup S_{ij}$.

$\phi(i|\Omega \setminus \{j\})$ and $\phi(j|\Omega \setminus \{i\})$ represent the individual contributions of units $i$ and $j$, respectively, when the perturbation units $i, j$ work individually. The individual contribution of perturbation unit $i$, when perturbation unit $j$ is absent, is given as follows.

$$\phi(i|\Omega \setminus \{j\}) = \sum_{S \subseteq \Omega \setminus \{i,j\}} \frac{|S|! \, (n - |S| - 2)!}{(n - 1)!} (v(S \cup \{i\}) - v(S)).$$

In this game, because the perturbation unit $j$ is always absent, we can consider there are only $n - 1$ players in the game. Consequently the set of players changes to $\Omega \setminus \{j\}$.

Similarly, the individual contribution of perturbation unit $j$, when perturbation unit $i$ is absent, is given as follows.

$$\phi(j|\Omega \setminus \{i\}) = \sum_{S \subseteq \Omega \setminus \{i,j\}} \frac{|S|! \, (n - |S| - 2)!}{(n - 1)!} (v(S \cup \{j\}) - v(S)).$$

In Section 1, the interaction between perturbation units $\delta_i, \delta_j$ is defined as the change of the importance $\phi_i$ of the $i$-th unit when the $j$-th unit $\delta_j$ is perturbed *w.r.t* the case when the $j$-th unit $\delta_j$ is not perturbed. If the perturbation $\delta_j$ on the $j$-th unit increases the importance $\phi_i$ of the $i$-th unit, then there is a positive interaction between $\delta_i$ and $\delta_j$. If the perturbation $\delta_j$ decreases the importance $\phi_i$, it indicates a negative interaction. Mathematically, this definition can be written as follows.

$$I'_{ij}(\delta) = \phi_{i,w/\,j} - \phi_{i,w/o\,j}, \tag{12}$$

where $\phi_{i,w/\,j}$ represents the importance of $\delta_i$, when $\delta_j$ is always present; $\phi_{i,w/o\,j}$ represents the importance of $\delta_i$, when $\delta_j$ is always absent. When perturbation unit $j$ is always present, the contribution of perturbation unit $i$ is given as follows.

$$\phi_{i,w/\,j} = \sum_{S \subseteq \Omega \setminus \{i,j\}} \frac{|S|! \, (n - |S| - 2)!}{(n - 1)!} (v(S \cup \{i, j\}) - v(S \cup \{j\})).$$

In this game, because the perturbation unit $j$ is always present, we can consider there are only $n - 1$ players.

When perturbation unit $j$ is always absent, the contribution of perturbation unit $i$ is given as follows.

$$\phi_{i,w/o\,j} = \sum_{S \subseteq \Omega \setminus \{i,j\}} \frac{|S|! \, (n - |S| - 2)!}{(n - 1)!} (v(S \cup \{i\}) - v(S)).$$

In this game, because the perturbation unit $j$ is always absent, we can consider there are only $n - 1$ players.

The interaction in Equation (11) is equal to the interaction in Equation (12), *i.e.*

$$I_{ij}(\delta) = I'_{ij}(\delta)$$

## E    PROOF OF PROPOSITION 1

To simplify the problem setting, we do not consider some tricks in adversarial attacking, such as gradient normalization and the clip operation. In multi-step attacking, the final perturbation generated after $t$ steps is given as follows.

$$\delta^t_{\text{multi}} \stackrel{\text{def}}{=} \alpha \sum_{t'=0}^{t-1} \nabla_x \ell(h(x + \delta^{t'}_{\text{multi}}), y),$$

where $\alpha$ represents the step size, and $\ell(h(x), y)$ is referred as the classification loss.

To simplify the notation, we use $g(x)$ to denote $\nabla_x \ell(h(x), y)$, *i.e.* $g(x) \overset{\text{def}}{=} \nabla_x \ell(h(x), y)$. Furthermore, we define the update of the perturbation with the multi-step attack at each step $t$ as follows.

$$\Delta x_{\text{multi}}^t \overset{\text{def}}{=} \alpha \cdot g(x + \delta_{\text{multi}}^{t-1}). \tag{13}$$

In this way, the perturbation can be written as follows.

$$\delta_{\text{multi}}^t = \Delta x_{\text{multi}}^1 + \Delta x_{\text{multi}}^2 + \cdots + \Delta x_{\text{multi}}^{t-1}. \tag{14}$$

**Lemma 1.** *Given the sample $x \in \mathbb{R}^n$ and the adversarial perturbation $\delta \in \mathbb{R}^n$, we use $\Omega = \{1, 2, \ldots, n\}$ to denote the set of all perturbation units. The score function is denoted by $v(S) = L(x + \delta^{(S)})$, where $\delta^{(S)}$ satisfies $\forall i \in S, \delta_i^{(S)} = \delta_i; \forall i \notin S, \delta_i^{(S)} = 0$. The Shapley interaction between perturbation units $a, b$ can be written as $I_{ab} = \delta_a H_{ab}(x)\delta_b + \hat{R}_2(\delta)$, where $H_{ab}(x) = \frac{\partial L(x)}{\partial x_a \partial x_b}$ represents the element of the Hessian matrix, and $\hat{R}_2(\delta)$ denotes terms with elements in $\delta$ of higher than the second order.*

*Proof.* The Shapley interaction between perturbation units $a, b$ is

$$I_{ab}(\delta) = \sum_{S \subseteq \Omega \backslash \{a,b\}} \frac{|S|!\,(n - |S|-2)!}{(n-1)!} [v(S \cup \{a,b\}) - v(S \cup \{b\}) - v(S \cup \{a\}) + v(S)],$$

where $v(S) = L(x + \delta^{(S)})$. Here, the classification loss can be approximated as $L(x + \delta) = L(x) + g^T(x)\delta + \frac{1}{2}\delta^T H(x)\delta + R_2(\delta)$ using Taylor series. Thus, $\forall S' \subseteq \Omega$,

$$v(S') = L(x) + \sum_{a \in S'} g_a(x)\delta_a + \frac{1}{2} \sum_{a,b \in S'} \delta_a H_{ab}(x)\delta_b^{(S')} + R_2^{S'}(\delta).$$

where $R_2^{S'}(\delta)$ denotes terms with elements in $\delta^{(S')}$ of higher than the second order.

In this way, the Shapley interaction $I_{ab}$ is given as

$$I_{ab}(\delta) = \sum_{S \subseteq \Omega \backslash \{a,b\}} \frac{|S|!\,(n - |S|-2)!}{(n-1)!}[v(S \cup \{a,b\}) - v(S \cup \{b\}) - v(S \cup \{a\}) + v(S)]$$

$$= \sum_{S \subseteq \Omega \backslash \{a,b\}} \frac{|S|!\,(n - |S|-2)!}{(n-1)!}\{[L(x)+$$

$$\sum_{a' \in S \cup \{a,b\}} g_{a'}(x)\delta_{a'} + \frac{1}{2}\sum_{a',b' \in S \cup \{a,b\}} \delta_{a'}H_{a'b'}(x)\delta_{b'} + R_2^{(S \cup \{a,b\})}(\delta)]$$

$$- [L(x) + \sum_{a' \in S \cup \{b\}} g_{a'}(x)\delta_{a'} + \frac{1}{2}\sum_{a',b' \in S \cup \{b\}} \delta_{a'}H_{a'b'}(x)\delta_{b'} + R_2^{(S \cup \{b\})}(\delta)]$$

$$- [L(x) + \sum_{a' \in S \cup \{a\}} g_{a'}(x)\delta_{a'} + \frac{1}{2}\sum_{a',b' \in S \cup \{a\}} \delta_{a'}H_{a'b'}(x)\delta_{b'} + R_2^{(S \cup \{a\})}(\delta)]$$

$$+ L(x) + \sum_{a' \in S} g_{a'}(x)\delta_{a'} + \frac{1}{2}\sum_{a',b' \in S} \delta_{a'}H_{a'b'}(x)\delta_{b'} + R_2^{(S)}(\delta)\}$$

$$= \sum_{S \subseteq \Omega \backslash \{a,b\}} \frac{|S|!\,(n - |S|-2)!}{(n-1)!}\{\delta_a H_{ab}(x)\delta_b\}$$

$$+ \underbrace{\sum_{S \subseteq \Omega \backslash \{a,b\}} \frac{|S|!\,(n - |S|-2)!}{(n-1)!}[R_2^{(S \cup \{a,b\})}(\delta)) - R_2^{(S \cup \{a\})}(\delta) - R_2^{(S \cup \{b\})}(\delta) + R_2^{(S)}(\delta)]}_{\hat{R}_2(\delta)}$$

$$= \left\{ \sum_{s=0}^{n-2} \sum_{\substack{S \subseteq \Omega \backslash \{a,b\}, \\ |S|=s}} \frac{s!\,(n - s - 2)!}{(n-1)!}[\delta_a H_{ab}(x)\delta_b] \right\} + \hat{R}_2(\delta)$$

$$= \left\{ \sum_{s=0}^{n-2} \frac{(n-2)!}{s!\,(n-s-2)!} \frac{s!\,(n-s-2)!}{(n-1)!} [\delta_a H_{ab}(x)\delta_b] \right\} + \hat{R}_2(\delta)$$
$$= \delta_a H_{ab}(x)\delta_b + \hat{R}_2(\delta),$$

where $\hat{R}_2(\delta)$ denotes terms with elements in $\delta$ of higher than the second order. $\qquad\square$

**Lemma 2.** *The update of the perturbation with the multi-step attack at step $t$ defined in Equation (13) can be written as $\Delta x_{multi}^t = \alpha\,[I + \alpha H(x)]^{t-1} g(x) + \hat{R}_1^t$, where $g(x) \overset{def}{=} \nabla_x \ell(h(x), y)$ represents the gradient, and $H(x) \overset{def}{=} \nabla_x^2 \ell(h(x), y)$ represents the Hessian matrix. $\hat{R}_1^t$ denotes terms with elements in $\delta_{multi}^{t-1}$ of higher than the first order.*

*Proof.* If $t = 1$, $\Delta x_{multi}^1 = \alpha \cdot g(x)$.

Let $\forall t' < t, \Delta x_{multi}^{t'} = \alpha\,[I + \alpha H(x)]^{t'-1} g(x) + \hat{R}_1^{t'}$, then we have

$$\Delta x_{multi}^t = \alpha \cdot g(x + \delta_{multi}^{t-1}) \quad // \quad \text{According to Equation (13)}$$
$$= \alpha \cdot g(x + \Delta x_{multi}^1 + \Delta x_{multi}^2 + \cdots + \Delta x_{multi}^{t-1}) \quad // \quad \text{According to Equation (14)}$$
$$= \alpha \cdot g\left( x + \alpha \left[I + [I+\alpha H(x)] + [I+\alpha H(x)]^2 + \cdots + [I+\alpha H(x)]^{t-2}\right] g(x) + \sum_{t'=1}^{t-1} \hat{R}_1^{t'} \right),$$

where $\hat{R}_1^{t'}$ denotes terms of elements in $\delta_{multi}^{t'-1}$ of higher than the first order.

Using the Taylor series, we get

$$\Delta x_{multi}^t = \alpha \cdot g(x) + \alpha^2 H(x) T(x) + \underbrace{\alpha H(x) \sum_{t'=1}^{t-1} \hat{R}_1^{t'} + R_1^{t-1}}_{\hat{R}_1^t}, \tag{15}$$

where $R_1^{t-1}$ denotes terms with elements $\delta_{multi}^{t-1}$ of higher than the first order. $T(x)$ in Equation (15) is given as follows.

$$T(x) = \left[I + [I + \alpha H(x)] + [I + \alpha H(x)]^2 + \cdots + [I + \alpha H(x)]^{t-2}\right] g(x). \tag{16}$$

Multiply $(I + \alpha H(x))$ on both sides of Equation (16), and we get

$$(I + \alpha H(x))T(x) = \alpha \cdot \left[[I + \alpha H(x)] + [I + \alpha H(x)]^2 + \cdots + [I + \alpha H(x)]^{t-1}\right] g(x). \tag{17}$$

Then, according to Equation (17) and Equation (16), we get

$$H(x)T(x) = \left[[I + \alpha H(x)]^{t-1} - I\right] g(x). \tag{18}$$

Substituting Equation (18) back to Equation (15), we have

$$\Delta x_{multi}^t = \alpha\,[I + \alpha H(x)]^{t-1} g(x) + \hat{R}_1^t.$$

In this way, we have proved that $\forall t \geq 1, \Delta x_{multi}^t = \alpha\,[I + \alpha H(x)]^{t-1} g(x) + \hat{R}_1^t.$ $\qquad\square$

**Proposition 1.** *The adversarial perturbation generated by the multi-step attack via gradient descent is given as $\delta_{multi}^m = \alpha \sum_{t=0}^{m-1} \nabla_x \ell(h(x + \delta_{multi}^t), y)$, where $\delta_{multi}^t$ denotes the perturbation after the t-th step of updating, and $m$ is referred to as the total number of steps. The adversarial perturbation generated by the single-step attack is given as $\delta_{single} = \alpha m \nabla_x \ell(h(x), y)$. The expectation of interactions between perturbation units in $\delta_{multi}^m$, $\mathbb{E}_{a,b}[I_{ab}(\delta_{multi}^m)]$, is larger than $\mathbb{E}_{a,b}[I_{ab}(\delta_{single})]$, i.e. $\mathbb{E}_{a,b}[I_{ab}(\delta_{multi}^m)] \geq \mathbb{E}_{a,b}[I_{ab}(\delta_{single})]$.*

### E.1 FAIRNESS OF COMPARISONS OF INTERACTIONS INSIDE DIFFERENT PERTURBATIONS

Proposition 1 is valid for different loss functions of generating of adversarial perturbations. In this section, we discuss the fairness of comparisons of interactions inside different perturbations.

When we compare interactions inside different perturbations, magnitudes of these perturbations should be similar, because the comparison of interactions between adversarial perturbations of different magnitudes is not fair. For fair comparisons, in Section 3.1, this paper controls the magnitude of the single-step attack by setting the step size of the single-step attack as $\alpha m$, where $\alpha$ and $m$ denotes the step size and the total number of steps of the multi-step attack, respectively. The equivalent step size $\alpha m$ makes the magnitude of perturbations generated by the single-step attack to be similar to that of perturbations generated by the multi-step attack, when we use the target score before the softmax layer to generate adversarial perturbations, such as $\tilde{\ell}(h(x), y) = \max_{y' \neq y} h(x) - h_y(x)$. In this case, the magnitude of the gradient $\nabla_x \tilde{\ell}(h(x), y)$ is relatively stable. In particular, this type of loss has been widely used. For example, one of the most widely used attacking (Carlini & Wagner, 2017), uses the score before the softmax layer for targeted attacking.

### E.2 PROOF OF PROPOSITION 1

*Proof.* According to Lemma 2, the update of the perturbation with the multi-step attack at the step $t$ is given as follows.

$$\Delta x_{\text{multi}}^t = \alpha \left[ I + \alpha H(x) \right]^{t-1} g(x) + \hat{R}_1^t, \tag{19}$$

where $\hat{R}_1^t$ denotes terms with elements in $\delta_{\text{multi}}^{t-1}$ of higher than the first order, and $\alpha$ represents the step size.

To simplify the notation without causing ambiguity, we write $g(x)$ and $H(x)$ as $g$ and $H$, respectively. In this way, according to Equation (14) and Equation (19), $\delta_{\text{multi}}^m$ can be written as follows.

$$\begin{aligned} \delta_{\text{multi}}^m &= \alpha \left[ I + [I + \alpha H] + [I + \alpha H]^2 + \cdots + [I + \alpha H]^{m-1} \right] g + \sum_{t=1}^{m} \hat{R}_1^t \\ &= \alpha \left[ mI + \frac{\alpha m(m-1)}{2} H + \dots \right] g + \sum_{t=1}^{m} \hat{R}_1^t, \end{aligned} \tag{20}$$

where $m$ represents the total number of steps. According to Lemma 1, the Shapley interaction between perturbation units $a, b$ in $\delta_{\text{multi}}^m$ is given as follows.

$$I_{ab}(\delta_{\text{multi}}^m) = \delta_{\text{multi},a}^m H_{ab} \delta_{\text{multi},b}^m + \hat{R}_2(\delta_{\text{multi}}^m), \tag{21}$$

where $\hat{R}_2(\delta_{\text{multi}}^m)$ denotes terms with elements in $\delta_{\text{multi}}^m$ of higher than the second order.

According to Equation (20) and Equation (21), we have

$$
I_{ab}(\delta_{\text{multi}}^m) = H_{ab}\Big[\alpha m g_a + \frac{\alpha^2 m(m-1)}{2}\sum_{b'=1}^n (H_{ab'}g_{b'}) + \cdots + \sum_{t=1}^m \underbrace{o(\delta_{\text{multi},a}^t)}_{\substack{\text{terms of }\delta_{\text{multi},a}^t \\ \text{of higher than the first order,} \\ \text{which corresponds to the term of} \\ \hat{R}_1^t \text{ in Equation (20)}}}\Big]\Big[
$$

$$
\alpha m g_b + \frac{\alpha^2 m(m-1)}{2}\sum_{a'=1}^n (H_{a'b}g_{a'}) + \cdots + \underbrace{o(\delta_{\text{multi},b}^t)}_{\substack{\text{terms of }\delta_{\text{multi},b}^t \\ \text{of higher than the first order,} \\ \text{which corresponds to the term of} \\ \hat{R}_1^t \text{ in Equation (20)}}}\Big] + \hat{R}_2(\delta_{\text{multi}}^m)
$$

$$
= \underbrace{\alpha^2 m^2 g_a g_b H_{ab}}_{\text{first-order terms }w.r.t.\text{ elements in }H}
$$

$$
+ \underbrace{\Big[\frac{\alpha^3(m-1)m^2}{2}g_b\sum_{b'=1}^n (H_{ab'}g_{b'}) + \frac{\alpha^3(m-1)m^2}{2}g_a\sum_{a'=1}^n (H_{a'b}g_{a'})\Big]H_{ab}}_{\text{second-order terms }w.r.t.\text{ elements in }H}
$$

$$
+ \underbrace{\Big[\frac{\alpha^4(m-1)^2 m^2}{4}\sum_{b'=1}^n (H_{ab'}g_{b'})\sum_{a'=1}^n (H_{a'b}g_{a'})H_{ab} + \dots\Big]}_{\mathcal{R}_2^{\text{multi}}(H)}
$$

$$
+ \underbrace{\big[\sum_{t=1}^m o(\delta_{\text{multi},a}^t)\big]H_{ab}\delta_{\text{multi},b}^m + \big[\sum_{t=1}^m o(\delta_{\text{multi},b}^t)\big]H_{ab}\delta_{\text{multi},a}^m + \hat{R}_2(\delta_{\text{multi}}^m)}_{\hat{R}_2'(\delta_{\text{multi}}^m)}
$$

$$
= \alpha^2 m^2 g_a g_b H_{ab} + \frac{\alpha^3(m-1)m^2}{2}g_a H_{ab}\sum_{a'=1}^n (H_{a'b}g_{a'})
$$

$$
+ \frac{\alpha^3(m-1)m^2}{2}g_b H_{ab}\sum_{b'=1}^n (H_{ab'}g_{b'}) + \hat{R}_2'(\delta_{\text{multi}}^m) + \mathcal{R}_2^{\text{multi}}(H),
$$

(22)

where $\mathcal{R}_2^{\text{multi}}(H)$ represents terms with elements in $H$ of higher than the second order, and $\hat{R}_2'(\delta_{\text{multi}}^m)$ represents terms with elements in $\delta_{\text{multi}}^m$ of higher than the second order.

Let us consider the single-step attack. When we compare interactions inside different perturbations, magnitudes of these perturbations should be similar, because the comparison of interactions between adversarial perturbations of different magnitudes is not fair. For fair comparisons, in Section 3.1, this paper controls the magnitude of the single-step attack, as follows. The single-step attack only uses the gradient information on the original input $x$, which generates adversarial perturbations as:

$$
\delta_{\text{single}} = \alpha m g.
$$

Therefore, according to Lemma 1, the interaction between perturbation units $a, b$ of $\delta_{\text{single}}$ is given as follows.

$$
\begin{aligned}
I_{ab}(\delta_{\text{single}}) &= \delta_{\text{single},a}H_{ab}\delta_{\text{single},b} + \hat{R}_2(\delta_{\text{single}}) \\
&= m^2\alpha^2 g_a g_b H_{ab} + \hat{R}_2(\delta_{\text{single}}),
\end{aligned}
$$

(23)

where $\hat{R}_2(\delta_{\text{single}})$ denotes terms with elements in $\delta_{\text{single}}$ of higher than the second order. In this way, according to Equation (22) and Equation (23), the expectation of the difference between $I_{ab}(\delta_{\text{multi}}^m)$ and $I_{ab}(\delta_{\text{single}})$ is given as follows.

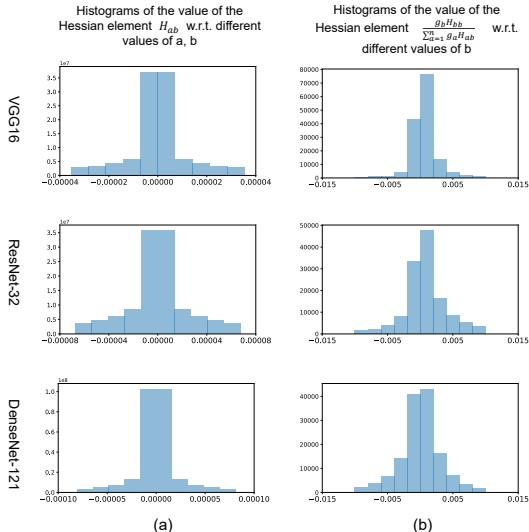

Figure 4: (a) Histograms of the value of the Hessian element $H_{ab}(x)$ *w.r.t.* different values of $a, b$. (b) Histograms of the value of $\frac{g_b H_{bb}}{\sum_{a=1}^{n} g_a H_{ab}}$ *w.r.t.* different values of $b$. Because the Hessian of the DNN with the ReLU activation is not well defined, we replace the ReLU activation with the Softplus activation $f(x) = \frac{1}{\beta} \log(1 + e^{-\beta x})$. We train VGG-16, ResNet-32, and DensetNet-121 on the CIFAR-10 dataset (Krizhevsky et al., 2009), and use the cross-entropy loss as the classification loss.

$$\mathbb{E}_{a,b} \left[ I_{ab}(\delta_{\text{multi}}^m) - I_{ab}(\delta_{\text{single}}) \right]$$

$$= \mathbb{E}_{a,b} \left[ \frac{\alpha^3(m-1)m^2}{2} g_a H_{ab} \sum_{a'=1}^{n} (H_{a'b} g_{a'}) + \frac{\alpha^3(m-1)m^2}{2} g_b H_{ab} \sum_{b'=1}^{n} (H_{ab'} g_{b'}) \right.$$

$$\left. + \hat{R}_2'(\delta_{\text{multi}}^m) + \mathcal{R}_2^{\text{multi}}(H) - \hat{R}_2(\delta_{\text{single}}) \right]$$

$$= \frac{\alpha^3(m-1)m^2}{2} \mathbb{E}_{a,b} \left[ \underbrace{g_a H_{ab} \sum_{a'=1}^{n} (H_{a'b} g_{a'})}_{U_{ab}} + \underbrace{g_b H_{ab} \sum_{b'=1}^{n} (H_{ab'} g_{b'})}_{U_{ba}} \right] + \mathbb{E}_{a,b} \left[ R_{ab} \right],$$

where

$$R_{ab} = \hat{R}_2'(\delta_{\text{multi}}^m) + \mathcal{R}_2^{\text{multi}}(H) - \hat{R}_2(\delta_{\text{single}}).$$

*Assumption 1*: Magnitudes of elements in the Hessian matrix $H(x)$ is small that $|H_{ab}(x)| \ll 1$, where $1 \leq a, b \leq n$. Therefore, $H^k(x) \approx 0$, if $k > 2$.

We verify the assumption by directly measuring the value of $H_{ab}(x)$. As Figure 4 (a) shows, the value of $H_{ab}(x)$ is very small that $|H_{ab}(x)| \ll 1$.

According to Assumption 1, we have $R_2^{\text{multi}}(H) \approx 0$. Note that the magnitude of $\delta_{\text{multi}}^m$ and the magnitude of $\delta_{\text{single}}$ are small, then $\hat{R}_2'(\delta_{\text{multi}}^m) \approx 0$, and $R_2(\delta_{\text{single}}) \approx 0$. In this way, we have $\mathbb{E}_{a,b}[R_{ab}] = \mathbb{E}_{a,b}[\hat{R}_2'(\delta_{\text{multi}}^m) + R_2(\delta_{\text{single}}) + R_2^{\text{multi}}(H)] \approx 0$.

Moreover, for the expectation of $U_{ab}$, we have

$$\mathbb{E}_{a,b}[U_{ab}] = \frac{1}{n(n-1)} \sum_{b=1}^{n} \sum_{a \neq b} g_a H_{ab} \sum_{a'=1}^{n} (g_{a'} H_{a'b})$$

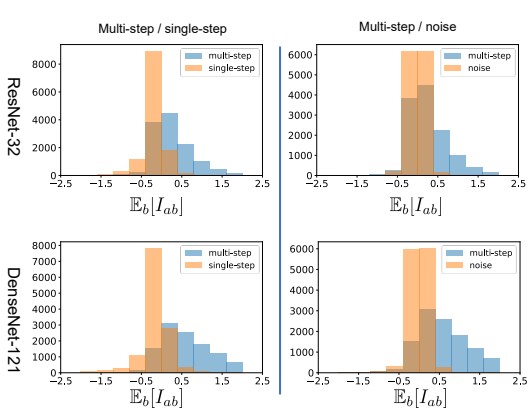

Figure 5: Histograms of the value of $\mathbb{E}_b[I_{ab}]$ *w.r.t.* different values of $a$

$$= \frac{1}{n(n-1)} \sum_{b=1}^{n} \left\{ \left[ \underbrace{\left( \sum_{a=1}^{n} g_a H_{ab} \right)}_{A} - \underbrace{g_b H_{bb}}_{B} \right] \underbrace{\left( \sum_{a'=1}^{n} g_{a'} H_{a'b} \right)}_{A} \right\}$$

Let us focus on terms of $A$ and $B$. Note that $A$ is the sum of $n$ terms ($n$ is large). In comparisons, $B$ is just a single term in $A$. Therefore, the sign of $A - B$ is usually dominated by the term $A$. In this way, we get $Prob\left[\text{sign}(A - B) = \text{sign}(A)\right] \approx 1$. Therefore, $Prob\left[(A - B)A \geq 0\right] \approx 1$. We verify this assumption by measuring the value of $\frac{g_b H_{bb}}{\sum_{a=1}^{n} g_a H_{ab}}$. If $Prob\left[|\frac{g_b H_{bb}}{\sum_{a=1}^{n} g_a H_{ab}}| \ll 1\right] \approx 1$, then we have $Prob\left[\text{sign}(A - B) = \text{sign}(A)\right] \approx 1$. As Figure 4 (b) shows, the value of $\frac{g_b H_{bb}}{\sum_{a=1}^{n} g_a H_{ab}}$ is very small that $|\frac{g_b H_{bb}}{\sum_{a=1}^{n} g_a H_{ab}}| \ll 1$. To this end, we have $(A - B)B \geq 0$, and we get

$$\mathbb{E}_{a,b}[U_{ab}] \geq 0 \tag{24}$$

Due to the symmetry of $a, b$, we have $\mathbb{E}_{a,b}[U_{ba}] = \mathbb{E}_{a,b}[U_{ab}]$. Therefore,

$$\mathbb{E}_{a,b}\left[I_{ab}(\delta_{\text{multi}}) - I_{ab}(\delta_{\text{single}})\right]$$
$$= \frac{\alpha^3 (m-1)m^2}{2} \mathbb{E}_{a,b}\left[U_{ab} + U_{ba}\right] + \mathbb{E}_{a,b}\left[R_{ab}\right]$$
$$\approx \alpha^3 (m-1)m^2 \mathbb{E}_{a,b}[U_{ab}] + 0$$
$$\geq 0.$$

$\square$

### E.3 VERIFICATION OF PROPOSITION 1

We verify that perturbations generated by the multi-step attack tend to exhibit larger interaction than those generated by the single-step attack by measuring the value of $\mathbb{E}_b[I_{ab}]$. As shown in Appendix F, we prove that $\mathbb{E}_b[I_{ab}] = v(\Omega) - v(\Omega \setminus \{a\}) - v(\{a\}) + v(\emptyset)$. Because the image data is high-dimensional, the cost of computing $\mathbb{E}_b[I_{ab}]$ is high. As Appendix J.1 demonstrates, given the input image, we can measure the interaction at the grid level, instead of the pixel level, to reduce the computational cost. Therefore, we divide the input image into $16 \times 16$ ($L = 16$) grids, and use Equation (39) to compute the interaction as $\mathbb{E}_{(p',q')}\left[I_{(p,q),(p'q')}(\delta)\right] = v(\Lambda) - v(\Lambda \setminus \{\Lambda_{pq}\}) - v(\{\Lambda_{pq}\}) + v(\emptyset)$, where $(p, q)$ denotes the coordinate of a grid. The experiments were conducted with ImageNet validation images on ResNet-32 and DenseNet-121.

For fair comparisons, the magnitude of perturbations generated by the single-step attack is controlled to be same as that generated by the multi-step attack. As Figure 5 (left) shows, perturbations generated by the multi-step attack tend to exhibit larger interaction than those generated by the single-step attack.

### E.4 PERTURBATIONS GENERATED BY THE MULTI-STEP ATTACK TEND TO EXHIBIT LARGER INTERACTION THAN GAUSSIAN NOISE

Moreover, we compare the interaction inside perturbation units generated by the multi-step attack with the Gaussian noise perturbation. Similarly, for fair comparisons, the magnitude of the Gaussian noise is controlled to be similar to that generated by the multi-step attack. As Figure 5 (right) shows, perturbations generated by the multi-step attack tend to exhibit larger interaction than Gaussian noise.

## F  EXPECTATION OF THE SHAPLEY INTERACTION

In Equation (3), the Shapley interaction between two perturbation units $i, j$ is given as follows.

$$I_{ij}(\delta) = \phi(S_{ij}|\Omega \setminus \{i,j\} \cup S_{ij}) - (\phi(i|\Omega \setminus \{j\}) + \phi(j|\Omega \setminus \{i\})),$$

where $\phi(S_{ij}|\Omega \setminus \{i,j\} \cup S_{ij})$ is the Shapley value of the singleton unit $S_{ij} = \{i, j\}$, when perturbation units $i, j$ form a coalition. $\phi(i|\Omega \setminus \{j\})$ and $\phi(j|\Omega \setminus \{i\})$ are Shapley values of perturbation units $i, j$, when these two perturbation units work individually. In this way, we can write the Shapley interaction in a closed form as follows.

$$I_{ij}(\delta) = \sum_{S \subseteq \Omega \setminus \{i,j\}} \frac{|S|! \, (n - |S| - 2)!}{(n-1)!} [v(S \cup \{i,j\}) - v(S \cup \{j\}) - v(S \cup \{i\}) + v(S)], \quad (25)$$

where $\forall S \subseteq \Omega, v(S) = \max_{y' \neq y} h_{y'}^{(s)}(x + \delta^{(S)}) - h_y^{(s)}(x + \delta^{(S)})$. The expectation of interaction is given as follows.

$$\mathbb{E}_{i,j}[I_{ij}(\delta)] = \frac{1}{n-1}\mathbb{E}_i[v(\Omega) - v(\Omega \setminus \{i\}) - v(\{i\}) + v(\emptyset)], \quad (26)$$

which is proved as follows.

*Proof.* As proved in Appendix D, $I_{ij}(\delta) = I'_{ij}(\delta)$. Therefore, the interaction between players $i$ and $j$ is given as follows.

$$I_{ij}(\delta) = \sum_{S \subseteq \Omega \setminus \{i,j\}} \frac{|S|! \, (n - |S| - 2)!}{(n-1)!} [[v(S \cup \{i,j\}) - v(S \cup \{j\})] - [v(S \cup \{i\}) - v(S)]]$$

$$= \phi_{j,w/\,i} - \phi_{j,w/o\,i}.$$

The expectation of the interaction can be written as follows.

$$\mathbb{E}_{i,j}[I_{ij}(\delta)] = \frac{1}{(n-1)}\mathbb{E}_i\left\{ \sum_{j \in \Omega \setminus \{i\}} [\phi_{j,w/\,i} - \phi_{j,w/o\,i}] \right\}.$$

According to the ***efficiency property*** of Shapley values (please refer to Appendix A.1 for details):

$$\sum_{j \in \Omega \setminus \{i\}} \phi_{j,w/\,i} = v(\Omega) - v\{i\}$$

$$\sum_{j \in \Omega \setminus \{i\}} \phi_{j,w/o\,i} = v(\Omega \setminus \{i\}) - v(\emptyset).$$

In this way,

$$\mathbb{E}_{i,j}[I_{ij}(\delta)] = \frac{1}{n-1}\mathbb{E}_i[v(\Omega) - v(\Omega \setminus \{i\}) - v(\{i\}) + v(\emptyset)].$$

$\square$

# G  DETAILS OF OBSERVING THE NEGATIVE CORRELATION BETWEEN THE TRANSFERABILITY AND THE INTERACTION

In Section 3.2, we directly measure the transfer utility and interactions of different adversarial perturbations. Here, we give more details of the experiments. We measure the transfer utility as *Transfer Utility* $= [\max_{y' \neq y} h_{y'}^{(t)}(x + \delta) - h_y^{(t)}(x + \delta)] - [\max_{y' \neq y} h_{y'}^{(t)}(x) - h_y^{(t)}(x)]$. We measure the interaction as $\mathbb{E}_{i,j}\left[I_{ij}(\delta)\right] = \frac{1}{n-1}\mathbb{E}_i\left[v(\Omega) - v(\Omega \setminus \{i\}) - v(\{i\}) + v(\emptyset)\right]$. As Appendix J.1 demonstrates, to reduce the computational cost, given the input image, we can measure the interaction at the grid level, instead of the pixel level. Therefore, we divide the input image into $16 \times 16$ ($L = 16$) grids, and use Equation (39) to compute the interaction as $\mathbb{E}_{(p,q),(p',q')}\left[I_{(p,q),(p'q')}(\delta)\right] = \frac{1}{L^2-1}\mathbb{E}_{(p,q)}\left[v(\Lambda) - v(\Lambda \setminus \{\Lambda_{pq}\}) - v(\{\Lambda_{pq}\}) + v(\emptyset)\right]$, where $(p,q)$ denotes the coordinate of a grid.

Using the validation set of the ImageNet dataset (Russakovsky et al., 2015), we generate adversarial perturbations on four types of DNNs, including ResNet-34/152(RN-34/152) (He et al., 2016) and DenseNet-121/201(DN-121/201) (Huang et al., 2017). We transfer adversarial perturbations generated on each ResNet to DenseNets. Similarly, we also transfer adversarial perturbations generated on each DenseNet to ResNets. Given an input image $x$, adversarial perturbations are generated using Equation (8), *i.e.* $\min_\delta -\ell(h(x + \delta), y) + c \cdot \|\delta\|_p^p$ s.t. $x + \delta \in [0, 1]^n$, where $c \in \mathbb{R}$ is a scalar constant. In this way, we gradually change the value of $c$ as different hyper-parameters to generate different adversarial perturbations, *i.e.* $c_k = k\beta + c_0$, where $\beta \in \mathbb{R}$ is a constant. Moreover, to ensure adversarial perturbations generated with different values of $c_k$ change smoothly, we use the perturbation generated with $c_{k-1}$ to initialize the perturbation for $c_k$, *i.e.* $\delta_{\text{init}}^{(c_k)} = \gamma\delta^{(c_{k-1})}$, where $\gamma \in \mathbb{R}$ is a constant. In our experiments, we set $\gamma = 0.6$. For fair comparisons, we need to ensure adversarial perturbations generated with different hyper-parameters $c$ to be comparable with each other. Thus, we select a constant $\tau$ and let $\|\delta\|_2 = \tau$ as the stopping criteria of all adversarial attacks. We set the number of steps as 1000. The threshold $\tau$ is set to ensure that attacks with different hyper-parameters $\tau$ are almost converged when the $L_2$ norm of the perturbation $\|\delta\|_2$ reaches $\tau$. Note that different attacking methods may successfully attack different sets of testing samples, so we select testing samples that can be successfully attacked by all attacking methods with different $c_k$ values (*i.e.* those having reached the stopping criteria under all attacks). The interaction and the transfer utility reported in Figure 1 are measured on the selected samples for fair comparisons.

# H  PROOF OF PROPOSITION 2

To simplify the problem setting, we do not consider some tricks in adversarial attacking, such as gradient normalization and the clip operation. In VR attack (Wu et al., 2018) , the final perturbation generated after $t$ steps is given as follows.

$$\delta_{\text{vr}}^t \overset{\text{def}}{=} \alpha \sum_{t'=0}^{t-1} \nabla_x \hat{\ell}(h(x + \delta_{\text{vr}}^{t'}), y),$$

where

$$\hat{\ell}(h(x), y) = \mathbb{E}_{\xi \sim \mathcal{N}(0,\sigma^2 I)}\left[\ell(h(x + \xi), y)\right]. \tag{27}$$

According to Equation (27), the gradient and the Hessian matrix of $\hat{\ell}(h(x), y)$ is given as follows.

$$
\begin{aligned}
\hat{g}(x) &= \nabla_x \hat{\ell}(h(x), y) \\
&= \mathbb{E}_{\xi \sim \mathcal{N}(0,\sigma^2 I)}\left[\nabla_x \ell(h(x + \xi), y)\right], \\
\hat{H}(x) &= \nabla_x^2 \hat{\ell}(h(x), y) \\
&= \mathbb{E}_{\xi \sim \mathcal{N}(0,\sigma^2 I)}\left[\nabla_x^2 \ell(h(x + \xi), y)\right].
\end{aligned}
\tag{28}
$$

where $\alpha$ represents the step size.

**Lemma 3.** *Given the Gaussian smoothed loss* $\hat{\ell}(x) = \mathbb{E}_{\xi \sim \mathcal{N}(0,\sigma^2 I)}\left[\ell(h(x), y)\right]$, *where* $\ell(h(x), y)$ *is the original classification loss,* $\forall a \neq b, \forall c \neq a$, *we have*

$$\mathbb{E}_x\left[\hat{g}_a^2(x)\hat{H}_{ab}^2(x) - g_a^2(x)H_{ab}^2(x)\right] \leq 0, \quad \mathbb{E}_x\left[\hat{g}_a(x)\hat{g}_b(x)\hat{H}_{ab}(x) - g_a(x)g_b(x)H_{ab}(x)\right] = 0,$$

$$\text{and } \mathbb{E}_x\left[\hat{g}_a(x)\hat{g}_c(x)\hat{H}_{ab}(x)\hat{H}_{cb}(x) - g_a(x)g_c(x)H_{ab}(x)H_{cb}(x)\right] = 0.$$

*Proof.* According to Equation (28), we have

$$\hat{g}_a(x) = \mathbb{E}_{\xi\sim\mathcal{N}(0,\sigma^2 I)}\left[g_a(x+\xi)\right] = \mathbb{E}_{x'\sim\mathcal{N}(x,\sigma^2 I)}\left[g_a(x')\right],$$

$$\hat{H}_{ab}(x) = \mathbb{E}_{\xi\sim\mathcal{N}(0,\sigma^2 I)}\left[\frac{\partial g_a(x+\xi)}{\partial x_b}\right] = \mathbb{E}_{x'\sim\mathcal{N}(x,\sigma^2 I)}\left[\frac{\partial g_a(x')}{\partial x_b}\right] = \mathbb{E}_{x'\sim\mathcal{N}(x,\sigma^2 I)}[H_{ab}(x')].$$

This indicates that the gradient and the Hessian matrix in the VR attack are both smoothed by the Gaussian noise. Because the Lipschitz constants of $g_a(x)$ and $H_{ab}(x)$ are usually limited to a certain range, we can ignore the tiny probability of large gradients and large elements in the Hessian matrix, and roughly assume that $g_a(x) \sim \mathcal{N}(\hat{g}_a(x), \sigma_{g_a}^2)$, and $H_{ab}(x) \sim \mathcal{N}(\hat{H}_{ab}(x), \sigma_{H_{ab}}^2)$, where $\sigma_{g_a}, \sigma_{H_{ab}} \in \mathbb{R}$ are tow constants denoting the standard deviation. Thus, $g_a(x)$ and $H_{ab}(x)$ can be written as follows.

$$g_a(x) = \hat{g}_a(x) + \epsilon_{g_a}, \quad \epsilon_{g_a} \sim \mathcal{N}(0, \sigma_{g_a}^2),$$

$$H_{ab}(x) = \hat{H}_{ab}(x) + \epsilon_{H_{ab}}, \quad \epsilon_{H_{ab}} \sim \mathcal{N}(0, \sigma_{H_{ab}}^2). \tag{29}$$

To simplify the notation without causing ambiguity, we write $\hat{g}(x)$ and $\hat{H}(x)$ as $\hat{g}$ and $\hat{H}$, respectively. Moreover, we write $g(x)$ and $H(x)$ as $g$ and $H$, respectively. In this way, we have

$$\mathbb{E}_x\left[\hat{g}_a^2\hat{H}_{ab}^2 - g_a^2 H_{ab}^2\right]$$

$$= \mathbb{E}_x\mathbb{E}_{\epsilon_{g_a},\epsilon_{H_{ab}}}\left[\hat{g}_a^2\hat{H}_{ab}^2 - (\hat{g}_a + \epsilon_{g_a})^2(\hat{H}_{ab} + \epsilon_{H_{ab}})^2\right]$$

$$= -\mathbb{E}_x\mathbb{E}_{\epsilon_{g_a},\epsilon_{H_{ab}}}\left[\epsilon_{g_a}^2\hat{H}_{ab}^2 + \epsilon_{H_{ab}}^2(\hat{g}_a + \epsilon_{g_a})^2 + 2\epsilon_{g_a}\hat{g}_a\hat{H}_{ab} + 2\epsilon_{H_{ab}}\hat{H}_{ab}(\hat{g}_a + \epsilon_{g_a})^2\right]$$

$$\leq -\mathbb{E}_x\mathbb{E}_{\epsilon_{g_a},\epsilon_{H_{ab}}}\left[2\epsilon_{g_a}\hat{g}_a\hat{H}_{ab} + 2\epsilon_{H_{ab}}\hat{H}_{ab}(\hat{g}_a + \epsilon_{g_a})^2\right]$$

$$= -\mathbb{E}_x\left\{\mathbb{E}_{\epsilon_{g_a}}\left[\epsilon_{g_a}\right]2\hat{g}_a\hat{H}_{ab} + \mathbb{E}_{\epsilon_{g_a}}\left[\mathbb{E}_{\epsilon_{H_{ab}}}[\epsilon_{H_{ab}}]2\hat{H}_{ab}(\hat{g}_a + \epsilon_{g_a})^2\right]\right\}$$

$$= -\mathbb{E}_x\left\{0\cdot 2\hat{g}_a\hat{H}_{ab} + \mathbb{E}_{\epsilon_{g_a}}\left[0\cdot 2\hat{H}_{ab}(\hat{g}_a + \epsilon_{g_a})^2\right]\right\}$$

$$= 0.$$

According to Equation (29), we have $g_a = \hat{g}_a + \epsilon_{g_a}$, $g_b = \hat{g}_b + \epsilon_{g_b}$. Thus, we have

$$\mathbb{E}_x\left[\hat{g}_a\hat{g}_b\hat{H}_{ab} - g_a g_b H_{ab}\right]$$

$$= \mathbb{E}_x\left[\hat{g}_a(x)\hat{g}_b\hat{H}_{ab} - (\hat{g}_a + \epsilon_{g_a})(\hat{g}_b + \epsilon_{g_b})(\hat{H}_{ab} + \epsilon_{H_{ab}})\right]$$

$$= -\mathbb{E}_{x,\epsilon_{g_a},\epsilon_{g_a},\epsilon_{H_{ab}}}\left[\epsilon_{g_a}\epsilon_{g_b}\epsilon_H + \epsilon_{g_a}\epsilon_{g_b}\hat{H}_{ab} + \epsilon_{g_b}\epsilon_{H_{ab}}\hat{g}_a\right.$$

$$\left. + \epsilon_{H_{ab}}\epsilon_{g_a}\hat{g}_b + \epsilon_{g_b}\hat{g}_b\hat{H}_{ab} + \epsilon_{g_b}\hat{g}_a\hat{H}_{ab} + \epsilon_{H_{ab}}\hat{g}_a\hat{g}_b\right]$$

$$= -\mathbb{E}_x\left[\mathbb{E}_{\epsilon_{g_a}}[\epsilon_{g_a}]\mathbb{E}_{\epsilon_{g_b}}[\epsilon_{g_b}]\mathbb{E}_{\epsilon_{H_{ab}}}[\epsilon_H] + \mathbb{E}_{\epsilon_{g_a}}[\epsilon_{g_a}]\mathbb{E}_{\epsilon_{g_b}}[\epsilon_{g_b}]\hat{H}_{ab} + \mathbb{E}_{\epsilon_{g_b}}[\epsilon_{g_b}]\mathbb{E}_{\epsilon_{H_{ab}}}[\epsilon_{H_{ab}}]\hat{g}_a\right.$$

$$\left. + \mathbb{E}_{\epsilon_{H_{ab}}}[\epsilon_{H_{ab}}]\mathbb{E}_{\epsilon_{g_a}}[\epsilon_{g_a}]\hat{g}_b + \mathbb{E}_{\epsilon_{g_a}}[\epsilon_{g_a}]\hat{g}_b\hat{H}_{ab} + \mathbb{E}_{\epsilon_{g_b}}[\epsilon_{g_b}]\hat{g}_a\hat{H}_{ab} + \mathbb{E}_{\epsilon_{H_{ab}}}[\epsilon_{H_{ab}}]\hat{g}_a\hat{g}_b\right]$$

$$= -\mathbb{E}_x\left[0\cdot 0\cdot 0 + 0\cdot 0\cdot\hat{H}_{ab} + 0\cdot 0\cdot\hat{g}_a + 0\cdot 0\cdot\hat{g}_b + 0\cdot\hat{g}_b\hat{H}_{ab} + 0\cdot\hat{g}_a\hat{H}_{ab} + 0\cdot\hat{g}_a\hat{g}_b\right]$$

$$= 0.$$

Moreover, according to Equation (29), we have

$$\mathbb{E}_x\left[\hat{g}_a\hat{g}_c\hat{H}_{ab}\hat{H}_{cb} - g_a g_c H_{ab}H_{cb}\right]$$

$$= \mathbb{E}_{x,\epsilon_{g_a},\epsilon_{g_c},\epsilon_{H_{ab}},\epsilon_{H_{cb}}} \left[ \hat{g}_a \hat{g}_c \hat{H}_{ab} \hat{H}_{cb} - (\hat{g}_a + \epsilon_{g_a})(\hat{g}_c + \epsilon_{g_c})(\hat{H}_{ab} + \epsilon_{H_{ab}})(\hat{H}_{cb} + \epsilon_{H_{cb}}) \right]$$

$$= -\mathbb{E}_{x,\epsilon_{g_a},\epsilon_{g_c},\epsilon_{H_{ab}},\epsilon_{H_{cb}}} \Big[ \epsilon_{g_a} \epsilon_{g_c} \epsilon_{H_{ab}} \epsilon_{H_{cb}}$$

$$+ \epsilon_{g_c} \epsilon_{H_{ab}} \epsilon_{H_{cb}} \hat{g}_a + \epsilon_{g_a} \epsilon_{H_{ab}} \epsilon_{H_{cb}} \hat{g}_c + \epsilon_{g_a} \epsilon_{g_c} \epsilon_{H_{cb}} \hat{H}_{ab} + \epsilon_{g_a} \epsilon_{g_c} \epsilon_{H_{ab}} \hat{H}_{cb}$$

$$+ \epsilon_{g_a} \epsilon_{g_c} \hat{H}_{ab} \hat{H}_{cb} + \epsilon_{g_a} \epsilon_{H_{ab}} \hat{g}_c \hat{H}_{cb} + \epsilon_{g_a} \epsilon_{H_{cb}} \hat{g}_c \hat{H}_{ab}$$

$$+ \epsilon_{g_c} \epsilon_{H_{ab}} \hat{g}_a \hat{H}_{cb} + \epsilon_{g_c} \epsilon_{H_{cb}} \hat{g}_a \hat{H}_{ab} + \epsilon_{H_{ab}} \epsilon_{H_{cb}} \hat{g}_a \hat{g}_c$$

$$+ \epsilon_{g_a} \hat{g}_c \hat{H}_{ab} \hat{H}_{cb} + \epsilon_{g_c} \hat{g}_a \hat{H}_{ab} \hat{H}_{cb} + \epsilon_{H_{ab}} \hat{g}_a \hat{g}_c \hat{H}_{cb} + \epsilon_{H_{cb}} \hat{g}_a \hat{g}_c \hat{H}_{ab} \Big]$$

$$= -\mathbb{E}_x \Big[ \mathbb{E}_{\epsilon_{g_a}}[\epsilon_{g_a}] \mathbb{E}_{\epsilon_{g_c}}[\epsilon_{g_c}] \mathbb{E}_{\epsilon_{H_{ab}}}[\epsilon_{H_{ab}}] \mathbb{E}_{\epsilon_{H_{cb}}}[\epsilon_{H_{cb}}] + \mathbb{E}_{\epsilon_{g_c}}[\epsilon_{g_c}] \mathbb{E}_{\epsilon_{H_{ab}}}[\epsilon_{H_{ab}}] \mathbb{E}_{\epsilon_{H_{cb}}}[\epsilon_{H_{cb}}] \hat{g}_a$$

$$+ \mathbb{E}_{\epsilon_{g_a}}[\epsilon_{g_a}] \mathbb{E}_{\epsilon_{H_{ab}}}[\epsilon_{H_{ab}}] \mathbb{E}_{\epsilon_{H_{cb}}}[\epsilon_{H_{cb}}] \hat{g}_c + \mathbb{E}_{\epsilon_{g_a}}[\epsilon_{g_a}] \mathbb{E}_{\epsilon_{g_c}}[\epsilon_{g_c}] \mathbb{E}_{\epsilon_{H_{cb}}}[\epsilon_{H_{cb}}] \hat{H}_{ab}$$

$$+ \mathbb{E}_{\epsilon_{g_a}}[\epsilon_{g_a}] \mathbb{E}_{\epsilon_{g_c}}[\epsilon_{g_c}] \mathbb{E}_{\epsilon_{H_{ab}}}[\epsilon_{H_{ab}}] \hat{H}_{cb}$$

$$+ \mathbb{E}_{\epsilon_{g_a}}[\epsilon_{g_a}] \mathbb{E}_{\epsilon_{g_c}}[\epsilon_{g_c}] \hat{H}_{ab} \hat{H}_{cb} + \mathbb{E}_{\epsilon_{g_a}}[\epsilon_{g_a}] \mathbb{E}_{\epsilon_{H_{ab}}} \epsilon_{H_{ab}} \hat{g}_c \hat{H}_{cb} + \mathbb{E}_{\epsilon_{g_a}}[\epsilon_{g_a}] \mathbb{E}_{\epsilon_{H_{cb}}}[\epsilon_{H_{cb}}] \hat{g}_c \hat{H}_{ab}$$

$$+ \mathbb{E}_{\epsilon_{g_c}}[\epsilon_{g_c}] \mathbb{E}_{\epsilon_{H_{ab}}}[\epsilon_{H_{ab}}] \hat{g}_a \hat{H}_{cb} + \mathbb{E}_{\epsilon_{g_c}}[\epsilon_{g_c}] \mathbb{E}_{\epsilon_{H_{cb}}}[\epsilon_{H_{cb}}] \hat{g}_a \hat{H}_{ab} + \mathbb{E}_{\epsilon_{H_{ab}}}[\epsilon_{H_{ab}}] \mathbb{E}_{\epsilon_{H_{cb}}}[\epsilon_{H_{cb}}] \hat{g}_a \hat{g}_c$$

$$+ \mathbb{E}_{\epsilon_{g_a}}[\epsilon_{g_a}] \hat{g}_c \hat{H}_{ab} \hat{H}_{cb} + \mathbb{E}_{\epsilon_{g_c}}[\epsilon_{g_c}] \hat{g}_a \hat{H}_{ab} \hat{H}_{cb} + \mathbb{E}_{\epsilon_{H_{ab}}}[\epsilon_{H_{ab}}] \hat{g}_a \hat{g}_c \hat{H}_{cb} + \mathbb{E}_{\epsilon_{H_{cb}}}[\epsilon_{H_{cb}}] \hat{g}_a \hat{g}_c \hat{H}_{ab} \Big]$$

$$= -\mathbb{E}_x \Big[ 0 + 0 \cdot \hat{g}_a + 0 \cdot \hat{g}_c + 0 \cdot \hat{H}_{ab} + 0 \cdot \hat{H}_{cb}$$

$$+ 0 \cdot \hat{H}_{ab} \hat{H}_{cb} + 0 \cdot \hat{g}_c \hat{H}_{cb} + 0 \cdot \hat{g}_c \hat{H}_{ab} + 0 \cdot \hat{g}_a \hat{H}_{cb} + 0 \cdot \hat{g}_a \hat{H}_{ab} + 0 \cdot \hat{g}_a \hat{g}_c$$

$$+ 0 \cdot \hat{g}_c \hat{H}_{ab} \hat{H}_{cb} + 0 \cdot \hat{g}_a \hat{H}_{ab} \hat{H}_{cb} + 0 \cdot \hat{g}_a \hat{g}_c \hat{H}_{cb} + 0 \cdot \hat{g}_a \hat{g}_c \hat{H}_{ab} \Big]$$

$$= 0.$$

$\square$

**Proposition 2.** *The adversarial perturbation generated by multi-step attack is denoted by $\delta_{multi}^m = \alpha \sum_{t=0}^{m-1} \nabla_x \ell(h(x + \delta_{multi}^t), y)$. The adversarial perturbation generated by VR Attack is denoted by $\delta_{vr}^m = \alpha \sum_{t=0}^{m-1} \nabla_x \hat{\ell}(h(x + \delta_{vr}^t), y)$, where $\hat{\ell}(h(x + \delta_{vr}^t), y) = \mathbb{E}_{\xi \sim \mathcal{N}(0,\sigma^2 I)} [\ell(h(x + \delta_{vr}^t + \xi), y)]$. Perturbation units of $\delta_{vr}^m$ tend to exhibit smaller interaction than $\delta_{multi}^m$, i.e. $\mathbb{E}_x \mathbb{E}_{a,b}[I_{ab}(\delta_{vr}^m)] \leq \mathbb{E}_x \mathbb{E}_{a,b}[I_{ab}(\delta_{multi}^m)]$.*

*Proof.* To simplify the notation without causing ambiguity, we write $\hat{g}(x)$ and $\hat{H}(x)$ as $\hat{g}$ and $\hat{H}$, respectively. Moreover, we write $g(x)$ and $H(x)$ as $g$ and $H$, respectively.

Just like the conclusion in Equation (22), we can write the interaction between $\delta_{vr,a}^m$ and $\delta_{vr,b}^m$ as follows.

$$I_{ab}(\delta_{vr}^m) = \alpha^2 m^2 \hat{g}_a \hat{g}_b \hat{H}_{ab} + \frac{\alpha^3 (m-1) m^2}{2} \hat{g}_a \hat{H}_{ab} \sum_{a'=1}^n (\hat{H}_{a'b} \hat{g}_{a'})$$

$$+ \frac{\alpha^3 (m-1) m^2}{2} \hat{g}_b \hat{H}_{ab} \sum_{b'=1}^n (\hat{H}_{ab'} \hat{g}_{b'}) + \hat{R}_2^{vr}(\delta_{vr}^m) + \mathcal{R}_2^{vr}(\hat{H}),$$

(30)

where $\alpha$ denotes the step size, and $m$ denotes the total number of steps. To enable fair comparisons, we use the same step size $\alpha$ and number of steps $m$ as multi-step attack to make the magnitude of $\delta_{vr}$ match the magnitude of $\delta_{multi}$. $\mathcal{R}_2^{vr}(\hat{H})$ represents terms with elements in $\hat{H}$ of higher than the second order, and $\hat{R}_2^{vr}(\delta_{vr}^m)$ represents terms with elements in $\delta_{vr}^m$ of higher than the second order.

In this way, according to Equation (30) and Equation (22), the expectation of the difference between $I_{ab}(\delta_{vr}^m)$ and $I_{ab}(\delta_{multi}^m)$ is given as follows.

$$\mathbb{E}_x \mathbb{E}_{a,b} [I_{ab}(\delta_{vr}) - I_{ab}(\delta_{multi})]$$

$$= \frac{\alpha^3 (m-1) m^2}{2} \mathbb{E}_{a,b} \left\{ \mathbb{E}_x \left[ \left[ \hat{g}_a^2 \hat{H}_{ab}^2 - g_a^2 H_{ab}^2 \right] + \left[ \hat{g}_b^2 \hat{H}_{ab}^2 - g_b^2 H_{ab}^2 \right] \right] \right\} + \mathbb{E}_{a,b} \mathbb{E}_x [R_{ab}^{vr}],$$

where

$$R_{ab}^{\text{vr}} = \frac{\alpha^3(m-1)m^2}{2}\left[\underbrace{\sum_{a'\in\{1,2,\ldots,n\}\backslash\{a\}}\left[(\hat{g}_a\hat{g}_{a'}\hat{H}_{ab}\hat{H}_{a'b} - g_ag_{a'}H_{ab}H_{a'b})\right]}_{V_{ab}}\right.$$

$$\left.+ \underbrace{\sum_{b'\in\{1,2,\ldots,n\}\backslash\{b\}}\left[(\hat{g}_b\hat{g}_{b'}\hat{H}_{ab}\hat{H}_{ab'} - g_bg_{b'}H_{ab}H_{ab'})\right]}_{V_{ba}}\right]$$

$$+ \alpha^2 m^2\left[\hat{g}_a\hat{g}_b\hat{H}_{ab} - g_ag_bH_{ab}\right] + \hat{R}_2^{\text{vr}}(\delta_{\text{vr}}^m) - \hat{R}_2'(\delta_{\text{multi}}^m) + \mathcal{R}_2^{\text{vr}}(H) - R_2^{\text{multi}}(H)$$

The expectation of $R_{ab}^{\text{vr}}$ is give as follows.

$$\mathbb{E}_x\mathbb{E}_{a,b}[R_{ab}^{\text{vr}}]$$
$$= \frac{\alpha^3(m-1)m^2}{2}\left\{\mathbb{E}_{a,b}\left[\mathbb{E}_x[V_{ab}] + \mathbb{E}_x[V_{ba}] + \frac{2}{\alpha(m-1)}\mathbb{E}_x\left[\hat{g}_a\hat{g}_b\hat{H}_{ab} - g_ag_bH_{ab}\right]\right]\right\}$$
$$+ \mathbb{E}_x\mathbb{E}_{a,b}[\hat{R}_2^{\text{vr}}(\delta_{\text{vr}}^m) - \hat{R}_2'(\delta_{\text{multi}}^m) + R_2^{\text{vr}}(H) - R_2^{\text{multi}}(H)] \approx 0.$$

According to Assumption 1, we have $R_2^{\text{vr}}(H) \approx 0$, and $R_2^{\text{multi}}(H) \approx 0$. Note that the magnitude of $\delta_{\text{mi}}^m$ and the magnitude of $\delta_{\text{multi}}$ are small, then $\hat{R}_2'(\delta_{\text{vr}}^m) \approx 0$, and $R_2(\delta_{\text{multi}}^m) \approx 0$. According to Lemma 3, we have $\mathbb{E}_x\left[\hat{g}_a\hat{g}_b\hat{H}_{ab} - g_ag_bH_{ab}\right] = 0$, $\mathbb{E}_x\left[(\hat{g}_a\hat{g}_{a'}\hat{H}_{ab}\hat{H}_{a'b} - g_ag_{a'}H_{ab}H_{a'b})\right] = 0$. Therefore, we get $\mathbb{E}_x[V_{ab}] = 0$. In this way, $\mathbb{E}_x\mathbb{E}_{a,b}[R_{ab}^{\text{vr}}] \approx 0$.

Furthermore, according to Lemma 3, we have $\mathbb{E}_x\left[\hat{g}_a^2\hat{H}_{ab}^2\right] - \mathbb{E}_x\left[g_a^2H_{ab}^2\right] \leq 0$.

Therefore,

$$\mathbb{E}_x\mathbb{E}_{a,b}\left[I_{ab}(\delta_{\text{vr}}) - I_{ab}(\delta_{\text{multi}})\right]$$
$$= \frac{\alpha^3(m-1)m^2}{2}\mathbb{E}_{a,b}\left\{\mathbb{E}_x\left[\hat{g}_a^2\hat{H}_{ab}^2 - g_a^2H_{ab}^2\right] + \mathbb{E}_x\left[\hat{g}_b^2\hat{H}_{ab}^2 - g_b^2H_{ab}^2\right]\right\} + \mathbb{E}_x\mathbb{E}_{a,b}[R_{ab}^{\text{vr}}]$$
$$\approx \frac{\alpha^3(m-1)m^2}{2}\mathbb{E}_{a,b}\left\{\mathbb{E}_x\left[\hat{g}_a^2\hat{H}_{ab}^2 - g_a^2H_{ab}^2\right] + \mathbb{E}_x\left[\hat{g}_b^2\hat{H}_{ab}^2 - g_b^2H_{ab}^2\right]\right\} + 0$$
$$\leq 0$$

$\square$

## I   PROOF OF PROPOSITION 3

To simplify the problem setting, we do not consider some tricks in adversarial attacking, such as gradient normalization and the clip operation. Note that the original MI Attack and the multi-step attack cannot be directly compared, since that magnitudes of the generated perturbations cannot be fairly controlled. The value of interactions is sensitive to the magnitude of perturbations. Comparing perturbations with different magnitudes is not fair. Thus, we slightly revise the MI Attack as

$$g_{\text{mi}}^t \overset{\text{def}}{=} \mu g_{\text{mi}}^{t-1} + (1-\mu)\nabla_x\ell(h(x+\delta_{\text{mi}}^{t-1}), y), \tag{31}$$

where $t$ denotes the step and $\mu = (t-1)/t$. $\ell(h(x), y)$ is referred as the classification loss. To simplify the notation, we use $g(x)$ to denote $\nabla_x\ell(h(x), y)$, i.e. $g(x) \overset{\text{def}}{=} \nabla_x\ell(h(x), y)$.

In MI attack, the final perturbation generated after $t$ steps is given as follows.

$$\delta_{\text{mi}}^t \overset{\text{def}}{=} \alpha\sum_{t'=0}^{t-1} g_{\text{mi}}^{t'},$$

where $\alpha$ represents the step size.

Furthermore, we define the update of perturbation with the MI attack at each step $t$ as follows.

$$\Delta x_{\text{mi}}^t \overset{\text{def}}{=} \alpha \cdot g_{\text{mi}}^t. \tag{32}$$

In this way, the perturbation can be written as follows.

$$\delta_{\text{mi}}^t = \Delta x_{\text{mi}}^1 + \Delta x_{\text{mi}}^2 + \cdots + \Delta x_{\text{mi}}^{t-1}. \tag{33}$$

**Lemma 4.** *The update of the perturbation with the MI attack at step $t$ defined in Equation (32) can be written as $\Delta x_{mi}^t = \alpha \left[ I + \alpha \frac{t-1}{2} H(x) + \mathcal{R}_1^t(H(x)) \right] g(x) + \tilde{R}_1^t$, where $\mathcal{R}_1^t(H(x))$ denotes terms of elements in $H(x)$ higher than the first order, and $\tilde{R}_1^t$ denotes terms with elements in $\delta_{mi}^{t-1}$ of higher than the first order.*

*Proof.* If $t = 1$, $\Delta x_{\text{mi}}^1 = \alpha \cdot g(x)$.

Let $\forall t' < t$, $\Delta x_{\text{mi}}^{t'} = \alpha \left[ I + \alpha \frac{t'-1}{2} H(x) + R_1^{t'}(H(x)) \right] g(x) + \tilde{R}_1^{t'}$.

According to Equation (31) and Equation (32), we have

$$\Delta x_{\text{mi}}^t = \alpha \cdot \left[ \frac{t-1}{t} g_{\text{mi}}^{t-1} + \frac{1}{t} g(x + \delta_{\text{mi}}^{t-1}) \right].$$

Applying the Taylor series to the term of $g(x + \delta_{\text{mi}}^{t-1})$, we get

$$\Delta x_{\text{mi}}^t = \alpha \cdot \left[ \frac{t-1}{t} g_{\text{mi}}^{t-1} + \frac{1}{t} \left[ g(x) + H(x)\delta_{\text{mi}}^{t-1} \right] + r_1^{t-1} \right], \tag{34}$$

where $r_1^{t-1}$ denotes terms of elements in $\delta_{\text{mi}}^{t-1}$ of higher than the first order.

According to Equation (33) and Equation (34), we get

$$\Delta x_{\text{mi}}^t = \alpha \cdot \left\{ \frac{t-1}{t} g_{\text{mi}}^{t-1} + \frac{1}{t} \left[ g(x) + H(x) \left[ \Delta x_{\text{mi}}^1 + \Delta x_{\text{mi}}^2 + \cdots + \Delta x_{\text{mi}}^{t-1} \right] \right] + r_1^{t-1} \right\}.$$

Because $\forall t' < t \ \Delta x_{\text{mi}}^{t'} = \alpha \left[ I + \alpha \frac{t'-1}{2} H(x) + \mathcal{R}_1^{t'}(H(x)) \right] g(x) + \tilde{R}_1^{t'}$, we have $\Delta x_{\text{mi}}^{t-1} = \alpha \cdot \left[ I + \alpha \frac{t-2}{2} H(x) + \mathcal{R}_1^{t-1}(H(x)) \right] g(x) + \tilde{R}_2^{t-1}$. According to Equation (32), we get $g_{\text{mi}}^{t-1} = \left[ I + \alpha \frac{t-2}{2} H(x) + \mathcal{R}_1^{t-1}(H(x)) \right] g(x) + \tilde{R}_2^{t-1}$.

In this way, we get

$$
\begin{aligned}
\Delta x_{\mathrm{mi}}^t = \alpha \cdot & \left\{ \frac{t-1}{t} \left[ I + \alpha \frac{t-2}{2} H(x) + R_1^{t-1}(H(x)) \right] g(x) \right. \\
& + \frac{1}{t} \left[ I + H(x) \left[ \alpha(t-1) + \alpha \frac{(t-2)(t-1)}{4} H(x) + \sum_{t'=1}^{t-1} \mathcal{R}_1^{t'}(H(x)) \right] g(x) + \sum_{t'=1}^{t-1} \tilde{R}_1^{t'} \right] \\
& \left. + r_1^{t-1} \right\}
\end{aligned}
$$

$$
= \alpha \cdot \left[ \underbrace{\frac{t-1}{t} \mathcal{R}_1^{t-1}(H(x)) + \frac{(t-2)(t-1)}{4t} H^2(x) + \frac{1}{t} H(x) \sum_{t'=1}^{t-1} \mathcal{R}_1^{t'}(H(x))}_{\mathcal{R}_1^t(H(x))} \right. 
$$

$$
\left. + I + \alpha \frac{t-1}{2} H(x) + \right] g(x) + \underbrace{\frac{1}{t} \sum_{t'=1}^{t-1} \tilde{R}_1^{t'} + r_1^{t-1}}_{\tilde{R}_1^t}
$$

$$
= \alpha \left[ I + \alpha \frac{t-1}{2} H(x) + \mathcal{R}_1^t(H(x)) \right] g(x) + \tilde{R}_1^t .
$$

where $\mathcal{R}_1^t(H(x))$ denotes terms of elements in $H(x)$ higher than the first order, and $\tilde{R}_1^t$ denotes terms with elements in $\delta_{\mathrm{mi}}^{t-1}$ of higher than the first order. In this way, we have proved that $\forall t \geq 1, \Delta x_{\mathrm{mi}}^t = \alpha \left[ I + \alpha \frac{t-1}{2} H(x) + \mathcal{R}_1^t(H(x)) \right] g(x) + \tilde{R}_1^t$. $\square$

**Proposition 3.** *The adversarial perturbation generated by multi-step attack is denoted by $\delta_{multi}^m = \alpha \sum_{t=0}^{m-1} \nabla_x \ell(h(x + \delta_{multi}^t), y)$. The adversarial perturbation generated by multi-step attack incorporating the momentum is computed as $\delta_{mi}^m = \alpha \sum_{t=0}^{m-1} g_{mi}^t$ Perturbation units of $\delta_{mi}^m$ exhibit smaller interactions than $\delta_{multi}^m$, i.e. $\mathbb{E}_{ij}[I_{ij}(\delta_{mi}^m)] \leq \mathbb{E}_{ij}[I_{ij}(\delta_{multi}^m)]$.*

*Proof.* According to Lemma 4, the update of the perturbation with the MI attack at the step $t$ is given as follows.

$$
\Delta x_{\mathrm{mi}}^t = \alpha \left[ I + \alpha \frac{t-1}{2} H(x) + \mathcal{R}_1^t(H(x)) \right] g(x) + \tilde{R}_1^t . \tag{35}
$$

where $\mathcal{R}_1^t(H(x))$ denotes terms of elements in $H(x)$ of higher than the first order. $\tilde{R}_1^t$ denotes terms with elements in $\delta_{\mathrm{mi}}^{t-1}$ of higher than the first order.

To simplify the notation without causing ambiguity, we write $g(x)$ and $H(x)$ as $g$ and $H$, respectively. In this way, according to Equation (33) and Equation (35), $\delta_{\mathrm{mi}}^m$ can be written as follows.

$$
\delta_{\mathrm{mi}}^m = \alpha \left[ mI + \frac{\alpha m(m-1)}{4} H + \sum_{t=1}^m \mathcal{R}_1^t(H) \right] g + \sum_{t=1}^m \tilde{R}_1^t . \tag{36}
$$

where $m$ represents the total number of steps. According to Lemma 1, the Shapley interaction between perturbation units $a, b$ in $\delta_{\mathrm{mi}}^m$ is given as follows.

$$
I_{ab}(\delta_{\mathrm{mi}}^m) = \delta_{\mathrm{mi},a}^m H_{ab} \delta_{\mathrm{mi},b}^m + \tilde{R}_2(\delta_{\mathrm{mi}}^m), \tag{37}
$$

where $\tilde{R}_2(\delta_{\mathrm{mi}}^m)$ denotes terms with elements in $\delta_{\mathrm{mi}}$ of higher than the second order.

According to Equation (36) and Equation (37), we get

$$
I_{ab}(\delta_{\mathrm{mi}}^m) = H_{ab}[\alpha m g_a + \frac{\alpha^2 m(m-1)}{4} \sum_{b'=1}^{n}(H_{ab'}g_{b'}) + \cdots + \sum_{t=1}^{m} \underbrace{o(\delta_{\mathrm{mi},a}^t)}_{\substack{\text{terms of } \delta_{\mathrm{mi},a}^t \\ \text{of higher than the first order,} \\ \text{which corresponds to the term of} \\ \tilde{R}_1^t \text{ in Equation (36)}}} ][
$$

$$
\alpha m g_b + \frac{\alpha^2 m(m-1)}{4} \sum_{a'=1}^{n}(H_{a'b}g_{a'}) + \cdots + \sum_{t=1}^{m} \underbrace{o(\delta_{\mathrm{mi},b}^t)}_{\substack{\text{terms of } \delta_{\mathrm{mi},b}^t \\ \text{of higher than the first order,} \\ \text{which corresponds to the term of} \\ \tilde{R}_1^t \text{ in Equation (36)}}} ] + \tilde{R}_2(\delta_{\mathrm{mi}}^m)
$$

$$
= \underbrace{\alpha^2 m^2 g_a g_b H_{ab}}_{\text{first-order terms } w.r.t. \text{ elements in } H}
$$

$$
+ \underbrace{\left[ \frac{\alpha^3(m-1)m^2}{4}g_b \sum_{b'=1}^{n}(H_{ab'}g_{b'}) + \frac{\alpha^3(m-1)m^2}{4}g_a \sum_{a'=1}^{n}(H_{a'b}g_{a'}) \right] H_{ab}}_{\text{second-order terms } w.r.t. \text{ elements in } H}
$$

$$
+ \underbrace{\left[ \frac{\alpha^4(m-1)^2 m^2}{16} \sum_{b'=1}^{n}(H_{ab'}g_{b'}) \sum_{a'=1}^{n}(H_{a'b}g_{a'})H_{ab} + \dots \right]}_{\mathcal{R}_2^{\mathrm{mi}}(H)}
$$

$$
+ \underbrace{[\sum_{t=1}^{m} o(\delta_{\mathrm{mi},a}^t)]H_{ab}\delta_{\mathrm{mi},b}^m + [o(\delta_{\mathrm{mi},b}^t)]H_{ab}\delta_{\mathrm{mi},a}^m + \tilde{R}_2(\delta_{\mathrm{mi}}^m)}_{\tilde{R}_2'(\delta_{\mathrm{mi}}^m)}
$$

$$
= \alpha^2 m^2 g_a g_b H_{ab} + \frac{\alpha^3(m-1)m^2}{4}g_a H_{ab}\sum_{a'=1}^{n}(H_{a'b}g_{a'})
$$

$$
+ \frac{\alpha^3(m-1)m^2}{4}g_b H_{ab}\sum_{b'=1}^{n}(H_{ab'}g_{b'}) + \tilde{R}_2'(\delta_{\mathrm{mi}}^m) + \mathcal{R}_2^{\mathrm{mi}}(H),
$$

$$(38)$$

where $\mathcal{R}_2^{\mathrm{mi}}(H)$ denotes terms of elements in $H$ higher than the second order, and $\tilde{R}_2'(\delta_{\mathrm{mi}}^m)$ denotes terms of elements in $\delta_{\mathrm{mi}}^m$ higher than the second order

According to Equation (22) and Equation (38), the expectation of the difference between $I_{ab}(\delta_{\mathrm{mi}}^m)$ and $I_{ab}(\delta_{\mathrm{multi}}^m)$ is given as follows.

$$
\mathbb{E}_{a,b}\left[ I_{ab}(\delta_{\mathrm{mi}}^m) - I_{ab}(\delta_{\mathrm{multi}}^m) \right]
$$

$$
= -\frac{\alpha^3(m-1)m^2}{4}\mathbb{E}_{a,b}\left[ \underbrace{g_a H_{ab}\sum_{a'=1}^{n}(H_{a'b}g_{a'})}_{U_{ab}} + \underbrace{g_b H_{ab}\sum_{b'=1}^{n}(H_{ab'}g_{b'})}_{U_{ba}} \right] + \mathbb{E}_{a,b}\left[ R_{ab}^{\mathrm{mi}} \right],
$$

where

$$
R_{ab}^{\mathrm{mi}} = \tilde{R}_2'(\delta_{\mathrm{mi}}^m) - \tilde{R}_2'(\delta_{\mathrm{multi}}^m) + \mathcal{R}_2^{\mathrm{mi}}(H) - \mathcal{R}_2^{\mathrm{multi}}(H).
$$

According to Assumption 1, we have $\mathcal{R}_2^{\mathrm{mi}}(H) \approx 0$, and $\mathcal{R}_2^{\mathrm{multi}}(H) \approx 0$. Note that the magnitude of $\delta_{\mathrm{mi}}^m$ and the magnitude of $\delta_{\mathrm{multi}}^m$ are small, then $\tilde{R}_2'(\delta_{\mathrm{mi}}^m) \approx 0$, and $R_2(\delta_{\mathrm{multi}}^m) \approx 0$. Therefore, $\mathbb{E}_{a,b}\left[ R_{ab}^{\mathrm{mi}} \right] = \mathbb{E}_{a,b}[\hat{R}_2'(\delta_{\mathrm{mi}}^m) - \tilde{R}_2'(\delta_{\mathrm{multi}}^m) + \mathcal{R}_2^{\mathrm{mi}}(H) - \mathcal{R}_2^{\mathrm{multi}}(H)] \approx 0$. Moreover, similar to Equation (24) in the proof of Proposition 1, we have $\mathbb{E}_{a,b}\left[ U_{ab} \right] = \mathbb{E}_{a,b}\left[ U_{ba} \right] \geq 0$.

Therefore,

$$\mathbb{E}_{a,b}\left[I_{ab}(\delta_{\mathrm{mi}}) - I_{ab}(\delta_{\mathrm{multi}})\right]$$

$$= -\frac{\alpha^3(m-1)m^2}{4}\mathbb{E}_{a,b}\left[U_{ab} + U_{ba}\right] + \mathbb{E}_{a,b}\left[R_{ab}^{\mathrm{mi}}\right]$$

$$\approx -\frac{\alpha^3(m-1)m^2}{2}\mathbb{E}_{a,b}\left[U_{ab}\right] + 0 \leq 0.$$

$\square$

Note that Proposition 3 just shows the revised MI Attack usually decreases the interaction between perturbation units. The proof towards all types of MI Attacks is still a challenge.

## J IMPLEMENTATION OF THE INTERACTION-REDUCED ATTACK (IR ATTACK)

### J.1 GRID-LEVEL INTERACTIONS FOR IMAGE DATA

Although the computation of $\mathbb{E}_{i,j}[I_{ij}(\delta)]$ can be simplified using Equation (26), the computational cost of $\mathbb{E}_{i,j}[I_{ij}(\delta)]$ is still high. Therefore, as Figure 6 shows, using the local property of images (Chen et al., 2018a), we can divide the entire image into $L \times L$ grids, and compute interactions at the grid level, instead of the pixel level. Let $\Lambda = \{\Lambda_{11}, \Lambda_{12}, \ldots, \Lambda_{LL}\}$ denote the set of grids. We use $(p,q)$ to denote the coordinate of a grid. In this way, the expectation of interactions between perturbation grids is given as follows.

$$\mathbb{E}_{(p,q),(p',q')}\left[I_{(p,q),(p'q')}(\delta)\right] = \frac{1}{L^2-1}\mathbb{E}_{(p,q)}\left[v(\Lambda) - v(\Lambda \setminus \{\Lambda_{pq}\}) - v(\{\Lambda_{pq}\}) + v(\emptyset)\right], \quad (39)$$

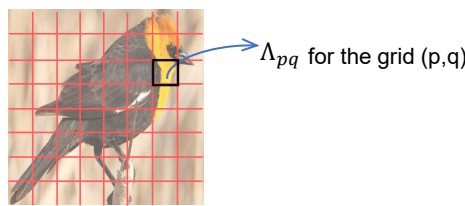

$\Lambda_{pq}$ for the grid (p,q)

Figure 6: For the input image, we can divide the image into grids, and compute interactions at the grid level.

### J.2 SCALABILITY OF THE INTERACTION LOSS

In this section, we discuss about two kinds of scalability of the interaction loss.

• **Is the computational cost of the interaction loss affordable when the number of players is large?** To this end, we have proved in Equation (4) that the computational complexity of the expectation of the interaction is linear, which is scalable. In fact, we do not directly compute interaction using Equation (3). Instead, we compute the expectation of interactions with Equation (4). The computational cost of the IR Attack can be further reduced by calculating the grid-level interactions of images.

We further conducted experiments to measure the time cost of generating perturbations using the IR Attack. We conducted the IR attack for 100 steps on the ImageNet dataset. The time cost was measured using PyTorch 1.6 (Paszke et al., 2019) on Ubuntu 18.04, with the Intel(R) Core(TM) i7-9800X CPU @ 3.80GHz and a Titan RTX GPU. Table 6 shows the average computational cost of generating adversarial perturbations on an input image with size $224 \times 224$ by the IR Attack for 100 steps. It shows that the IR Attack is computationally applicable to high-dimensional data and deep neural networks.

Table 6: Average computational cost of generating adversarial perturbations over an input image by the IR Attack for 100 steps on different source DNNs.

|  | RN-34 | RN-152 | DN-121 | DN-201 |
|---|---|---|---|---|
| Time (seconds) | 12.882 | 48.774 | 27.519 | 44.812 |

• **Is the computation cost of the interaction loss affordable when we consider the continuous space of adversarial perturbations?** It has been widely discussed (Ancona et al., 2019; Sundararajan & Najmi, 2019) that when applying the Shapley value, the feature space is regarded as binary. It is because as (Sundararajan & Najmi, 2019) shows that although there exist the Shapley-value-like attribution in a continuous space, only the Shapley value in the binary space is the unique attribution that satisfies *the linearity axiom, the dummy axiom, the symmetry axiom, and the efficiency axiom* that only in the binary space. Thus, when we compute the interaction, the perturbation can be regarded in the binary space, i.e., whether the perturbation unit is added to the input or not, which enables scalability.

## K    EVALUATION OF THE TRANSFERABILITY VIA LEAVE-ONE-OUT VALIDATION

As Figure 7 shows, the highest transferability of the MI Attack is achieved in an intermediate step, rather than in the last step. This phenomenon presents a challenge for fair comparisons of the transferability between different attacking methods.

To this end, in order to enable fair comparisons of transferability between different methods, we estimate the adversarial perturbations with the highest transferability for each input image via the leave-one-out (LOO) validation as follows. Given a set of clean examples $\{(x_i, y_i)\}_{i=1}^{N}$, where $y_i \in \{1, 2, \ldots, C\}$, we use $x_i^t$ to denote the adversarial example at step $t$ *w.r.t.* the clean example $x_i$, where $t \in \{1, 2, \ldots, T\}$, and $T$ is the number of total step. Given a target DNN $h(\cdot)$ and an input example $x$, where $h(\cdot)$ denotes the output before the softmax layer, we use $\mathcal{C}(x) = \arg\max_k h_k(x), k \in \{1, \ldots, C\}$ to denote the prediction of the example $x$.

$$\hat{x}_i \stackrel{\text{def}}{=} x_i^{t_i^*}, \quad s.t. \quad t_i^* = \arg\max_t \mathbb{E}_{i' \in \{1,2,\ldots,N\}\setminus\{i\}} \left[ \mathbb{1}[\mathcal{C}(x_{i'}^t) \neq y_{i'}] \right],$$

where $\mathbb{1}[\cdot]$ is the indicator function. Then the average transferability is given as follows.

$$Transferability = \mathbb{E}_i \left[ \mathbb{1}[\mathcal{C}(\hat{x}_i) \neq y_i] \right].$$

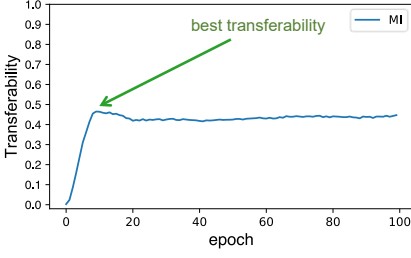

Figure 7: The curve of transferability in different steps.

## L    ADDITIONAL RELATED WORK

Some studies paid attention to intermediate features to improve transferability. Activation Attack (Inkawhich et al., 2019) forced the intermediate features of the input image to be similar with the intermediate features of a target image, in order to generate highly transferable targeted example. Distribution Attack (Inkawhich et al., 2020) explicitly modeled the feature distribution of each class, and improve the targeted transferability by driving the feature of perturbed input image into the distribution of a specific target class. Intermediate Level Attack (Huang et al., 2019) improved the

transferability of an adversarial example by maximizing the feature perturbation of a pre-specified layer. In comparison, we explain and improve the transferability based on game theory. Moreover, we discover the negative correlation between the transferability and interactions.

# M  ADDITIONAL EXPERIMENTS ON INTERACTION-REDUCED LOSS

## M.1  INTERACTION REDUCTION ON OTHER ATTACKS

To further demonstrate the effectiveness of the interaction loss, we have applied the interaction loss on other attacks besides the PGD Attack, including the MI Attack, the SGM Attack, and the VR Attack. More specifically, we added the interaction loss on the MI Attack (namely the MI+IR Attack), the SGM Attack (namely the SGM+IR Attack), and the VR Attack (namely the VR+IR Attack), respectively.

For the MI Attack and the SGM Attack, we directly applied Equation (7) to these attacks, because these attacks were compatible with the interaction loss. Besides, for the VR Attack, its objective function is given as follows.

$$\underset{\delta}{\text{maximize}} \quad \mathbb{E}_{\xi \sim \mathcal{N}(0,\sigma^2)} \left[ \ell(h(x + \delta + \xi), y) \right] \quad \text{s.t.} \quad \|\delta\|_p \leq \epsilon, \; x + \delta \in [0, 1]^n, \tag{40}$$

Therefore, the VR+IR Attack was implemented via sampling as follows.

$$\underset{\delta}{\text{maximize}} \quad \frac{1}{K} \sum_{k=1}^{K} \left[ \ell(h(x + \delta + \xi_k), y) - \mathbb{E}_{ij} \left[ I_{ij}(\delta) \right] \right], \quad \xi_k \sim \mathcal{N}\left(0, \sigma^2\right) \tag{41}$$
$$\text{s.t.} \quad \|\delta\|_p \leq \epsilon, \; x + \delta \in [0, 1]^n,$$

where the interaction loss was computed by considering the input image as $x + \xi_k$, rather than $x$ in Equation (26). The VR Attack reported in Table 4 followed the original paper (Wu et al., 2018) to set $K = 20$. However, a crucial issue for applying the interaction loss to the VR attack was its extremely high computational cost. Therefore, for the implementation of the VR+IR Attack, we set $K = 5$ and reduce the number of steps from 100 to 50. Just like experiments in Table 1, we also used the LOO strategy for evaluation.

Table 7, Table 8, and Table 9 compare the success rates of attacks with and without the interaction loss. The results demonstrated that the performance of the MI Attack, the SGM Attack, and the VR Attack can be further enhanced by directly adding the interaction loss to reduce interactions inside perturbations.

Table 7: The success rates of $L_\infty$ black-box attacks crafted by MI and MI+IR on four source models (RN-34/152, DN-121/201) against seven target models. The interaction loss can boost the transferability of MI.

| Source | Method | VGG-16 | RN152 | DN-201 | SE-154 | IncV3 | IncV4 | IncResV2 |
|--------|--------|--------|-------|--------|--------|-------|-------|----------|
| RN-34  | MI     | 80.1±0.5 | 73.0±2.3 | 77.7±0.5 | 48.9±0.8 | 46.2±1.2 | 39.9±0.5 | 34.8±2.5 |
|        | MI+IR  | **90.0±0.5** | **85.7±0.3** | **88.5±0.6** | **67.0±0.1** | **66.9±1.8** | **60.2±0.7** | **53.9±2.3** |
| RN-152 | MI     | 70.3±0.6 | – | 74.8±1.4 | 51.7±0.8 | 47.1±0.9 | 40.5±1.6 | 36.8±2.7 |
|        | MI+IR  | **78.9±1.4** | – | **82.2±2.0** | **68.3±0.3** | **63.6±1.2** | **59.0±0.4** | **56.3±1.0** |
| DN-121 | MI     | 83.0±4.9 | 72.0±0.7 | 91.5±0.2 | 58.4±2.6 | 54.6±1.6 | 49.2±2.4 | 43.9±1.5 |
|        | MI+IR  | **89.0±0.8** | **83.2±1.5** | **93.4±0.6** | **74.2±0.7** | **69.6±0.9** | **64.7±0.5** | **58.2±2.3** |
| DN-201 | MI     | 77.3±0.8 | 74.8±1.4 | – | 64.6±1.0 | 56.5±2.5 | 51.1±2.1 | 47.8±1.9 |
|        | MI+IR  | **87.3±0.3** | **81.6±2.0** | – | **75.4±0.6** | **66.6±3.3** | **60.0±1.0** | **62.1±0.7** |

## M.2  ATTACKS ON THE CIFAR-10 DATASET

In Table 1, we have shown that reduction of interactions could improve the adversarial transferability on the ImageNet dataset (Russakovsky et al., 2015). To further demonstrate the broad applicability of such a negative correlation, we also conducted the targeted attack on the CIFAR-10 dataset (Krizhevsky & Hinton, 2009) to test the transferability of perturbations generated with the interaction loss.

Table 8: The success rates of $L_\infty$ black-box attacks crafted by SGM and SGM+IR on four source models (RN-34/152, DN-121/201) against seven target models. The interaction loss can boost the transferability of SGM.

| Source | Method | VGG-16 | RN152 | DN-201 | SE-154 | IncV3 | IncV4 | IncResV2 |
|--------|--------|--------|-------|--------|--------|-------|-------|----------|
| RN-34 | SGM | 91.8±0.6 | 89.0±0.9 | 90.0±0.4 | 68.0±1.4 | 63.9±0.3 | 58.2±1.1 | 54.6±1.2 |
| | SGM+IR | **94.7±0.6** | **91.7±0.6** | **93.4±0.8** | **72.7±0.4** | **68.9±0.9** | **64.1±1.3** | **61.3±1.0** |
| RN-152 | SGM | 88.2±0.5 | – | 90.2±0.3 | 72.7±1.4 | 63.2±0.7 | 59.1±1.5 | 58.1±1.2 |
| | SGM+IR | **92.0±1.0** | – | **92.5±0.4** | **79.3±0.1** | **69.6±0.8** | **66.2±1.0** | **63.6±0.9** |
| DN-121 | SGM | 88.7±0.9 | 88.1±1.0 | 98.0±0.4 | 78.0±0.9 | 64.7±2.5 | 65.4±2.3 | 59.7±1.7 |
| | SGM+IR | **91.7±0.2** | **90.4±0.4** | **94.3±0.1** | **87.0±0.4** | **78.8±1.3** | **79.5±0.2** | **75.8±2.7** |
| DN-201 | SGM | 87.3±0.3 | 92.4±1.0 | – | 82.9±0.2 | 72.3±0.3 | 71.3±0.6 | 68.8±0.5 |
| | SGM+IR | **89.5±0.9** | **91.8±0.7** | – | **87.3±1.2** | **82.5±0.8** | **80.3±0.3** | **81.5±0.5** |

Table 9: The success rates of $L_\infty$ black-box attacks crafted by VR and VR+IR on four source models (RN-34/152, DN-121/201) against seven target models. The interaction loss can boost the transferability of VR.

| Source | Method | VGG-16 | RN152 | DN-201 | SE-154 | IncV3 | IncV4 | IncResV2 |
|--------|--------|--------|-------|--------|--------|-------|-------|----------|
| RN-34 | VR | 85.1 | 85.3 | 87.0 | 55.7 | 54.3 | 50.7 | 43.7 |
| | VR+IR | **90.8** | **92.2** | **93.3** | **75.4** | **75.4** | **67.5** | **66.1** |
| DN-121 | VR | 88.8 | 88.4 | **98.2** | 72.9 | 73.5 | 72.5 | 63.6 |
| | VR+IR | **93.0** | **93.5** | 96.2 | **83.7** | **82.8** | **84.0** | **79.8** |

Following Wu et al. (2018), we chose three DNNs as the source DNN or the target DNN, which included: LeNet (LeCun et al., 1998), RN-20 (He et al., 2016), and DN-121 (Huang et al., 2017). We conducted the targeted attack under the $L_\infty$ norm constraint, and chose he *plane* class as the target category. The norm constraint $\epsilon$ was set to 16/255, and the step size was set to 2/255. The transferability was computed based on the best adversarial perturbation during 50 steps via the leave-one-out (LOO) validation., which has been introduced in Appendix K.

As Table 10 shows, the transferability could be enhanced by reducing interactions on the targeted attack on the CIFAR-10 dataset. Particularly, when the source DNN is RN-20 and the target DNN is DN-121, the transferability improvement was about 30%, which was a considerable gain.

Table 10: The success rates of $L_\infty$ targeted black-box attacks on three source models, including LeNet, RN-20, DN-121, against three target models.

| Source | Method | LeNet | RN-20 | DN-121 |
|--------|--------|-------|-------|--------|
| LeNet | PGD | – | 34.1±0.1 | 19.6±0.4 |
| | PGD+IR | – | **44.2±0.3** | **29.7±1.1** |
| RN-20 | PGD | 10.8±0.8 | – | 41.9±1.3 |
| | PGD+IR | **19.7±0.3** | – | **71.8±1.0** |
| DN-121 | PGD | 10.0±0.7 | 44.2±0.3 | – |
| | PGD+IR | **18.9±0.7** | **58.5±1.0** | – |

Table 11: The average interaction inside adversarial perturbations generated by PGD, DI and TI.

| Method | RN-34 | DN-121 |
|--------|-------|--------|
| Baseline (PGD Attack) | **0.422** | **0.926** |
| DI Attack | 0.241 | 0.499 |
| TI Attack | 0.379 | 0.618 |

# N EMPIRICAL VERIFICATION OF OTHER TRANSFERABILITY-BOOSTING ATTACKS

We have theoretically analyzed the MI Attack, the VR Attack, and the SGM Attack. However, for other methods of improving adversarial transferability, such as Diversity Input (DI) (Xie et al.,

2019), which uses random data augmentation during attacking, it is difficult to mathematically prove that they essentially reduce interactions. Nevertheless, as Table 11 shows, we empirically demonstrated that two widely-used transferability-boosting attacks, DI and TI (Dong et al., 2019), also reduced interactions.

## O    ADDITIONAL EXPERIMENTS ON EFFECTS OF THE INTERACTION LOSS

We conducted additional experiments to test the effects of the interaction loss. We conducted attacks on two source DNNs (RN-34, DN-121), and transferred adversarial perturbations to seven target DNNs (VGG16, RN-152, DN-201, SE-154, IncV3, IncV4, IncResV2).

We used the following two experimental settings to compare the transferability of adversarial perturbations generated with different $\lambda$ values.

First, we re-drew the curves in Figure 3(a) by extending the $\lambda$ from the range of $[0, 1.2]$ to the range of $[0, 2.0]$, in order to show the performance of different $\lambda$ values. We simply changed the $\lambda$ value in the objective function (i.e. Equation (5)) without any other revisions. This was the most direct way to test the effects of $\lambda$. Experimental results are shown in Figure 8.

Besides above experimental settings, we also compared adversarial perturbations generated with different $\lambda$ values, when we controlled each perturbation to have the same attacking utility. The attacking utility was defined as follows.

$$Attacking\,Utility = \max_{y' \neq y} h_{y'}(x + \delta) - h_y(x + \delta)$$

where $y$ denote the label of the input image $x$. This setting also ensured the fairness of comparisons from a new perspective. Please see Figure 9 for experimental results.

In sum, under both experimental settings, we found that the large $\lambda$ value usually yielded a high adversarial transferability in our experiments.

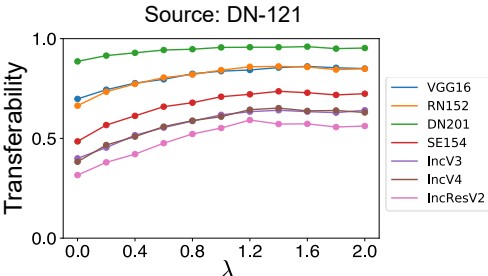

Figure 8: The success rates of black-box attacks with the IR Attack using different values of $\lambda$ under the first experimental setting.

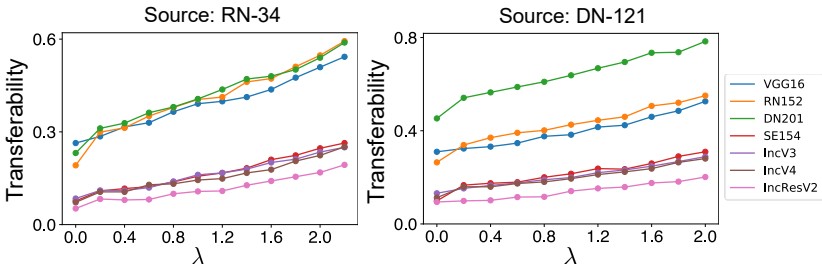

Figure 9: The success rates of black-box attacks with the IR Attack using different values of $\lambda$ under the second experimental setting.

