# OpenReview forum: "A Unified Approach to Interpreting and Boosting Adversarial Transferability"
_ICLR.cc/2021/Conference — ICLR 2021 Poster_

### Official Review · AnonReviewer4 · 2020-10-28
**Official Blind Review #4**

**Rating:** 6
**Confidence:** 3

**Review:**

Summary
The authors analyze the transferability of adversarial examples from the perspective of interactions based on game theory. They have discovered and shown  the negative correlation between the transferability and interactions inside adversarial perturbations. This discovery leads to an explanation of the adversarial transferability by the interaction inside adversarial perturbations. Thus, they proposed a new loss called interaction loss to penalize the interactions between perturbation units during attacking and experiments show the improvement of the adversarial transferability.

Strengths
1 An interesting understanding of adversarial transferability is provided.
2 Mathematical proofs are admired.
3 Results are very nicely presented, and writing is clear throughout the paper.

Weaknesses
1 Results are reported only on one dataset (ImageNet).
2 The appendix needs to be revised for better presentation.
3 The variances of the results are not reported.

This paper is well motivated because the authors observe the negative correlation between the transferability and interactions inside adversarial perturbations, and they provide a possible explanation of why the related research tasks  can improve the adversarial transferability. Further mathematical proofs and experiment results verify the observation.

However, I have some questions about this paper.

Questions
1.	There are many outliers off the blue shade in the subgraph of Figure 1. Could the authors give some interpretation of why there are so many outliers?

2.	For Equation 4, the value of the expected interactions is equivalent to the expectation of the contribution for each pixel. The authors aim to minimize the value of the expected interactions that is the same to average the contribution to all pixels. That may contradict the idea of the one-pixel attack. More interpretation should be given to understand the concept of interactions.

3.	The authors claim to provide a unified view to understand the enhancement of transferability; however the authors only explain three baseline models. Other types of adversarial example generation methods are not considered, such as Translation-Invariant Attack that the authors already mentioned in the related work. What is more, the Translation-Invariant Attack aims to compose the gradients of neighboring pixels together, which is contradictory to the authors’ idea of reducing the interaction of perturbations.


4.	From the experiment results in Table 1, why the performance of the IR attack is worse than the baseline when attacking RN-152 and DN-201 in the ensemble setting?

5.	In Section 5, Experiments, baselines: “the transferability of each baseline was computed based on the best adversarial perturbation during the 100 steps…” and you also mention that “Previous studies usually set the number of steps to 10 or 20”. I am confused about why you set the step to be 100? For a fair comparison, you should follow the setting in previous studies.

6.	In Section 5, Experiments, baselines: “we set $\lambda = 1$ for the IR Attack, when the source DNN was ResNet, and set $\lambda = 2$, for other source DNNs.”, but in Figure 3, part a, the range of $\lambda$ is from 0 to 1.2 for DN121, which does not include the \lamda value the authors suggested.

7.	The computational cost is not discussed. As the authors said in Section 5 “the computational cost of the interaction loss is intolerable”. No discussion about the running time is provided. Moreover, the authors choose the step to be 100, which further increases the running time.

8.	The third line on page 5, the term $|\delta|_{p}^{p}$ is wrong.

        It should be $|\delta |_{p}$.

9.	Appendix: The is some problem with the reference format about Figure 4 and Figure 7. I also suggest reorganizing the appendix.

---

> ### Author Response · Authors · 2020-11-19
> **Responses to Reviewer #4 (Part 1)**
>
> Thank you very much for your careful review and constructive comments. We try our best to answer all your concerns.
>
> Q1: "Results are reported only on one dataset (ImageNet)"
>
> A: Thank you. We have followed your suggestions to conduct **new experiments** on the CIFAR-10 dataset. Experimental results also proved that the adversarial transferability could be enhanced via reducing interactions. Please see Table 11 in Appendix N.3 for details.
> - - -
> Q2: **About paper writing.** "The appendix needs to be revised for better presentation. There is some problem with the reference format about Figure 4 and Figure 7. I also suggest reorganizing the appendix."
>
> A: Thank you for your careful review. We have followed your suggestion to revise the appendix. We fix the reference format problem of Figure 4 and Figure 7 in the appendix. We merge previous sections Appendix G and Appendix I into Appendix J in this version for better presentation.
> - - -
> Q3: "The variances of the results are not reported."
>
> A: We have followed your suggestions to conduct **new experiments** with different random samplings of grids or different initial perturbations to compute the variance. The variance is reported in Tables 1-4.
> - - -
> Q4: "There are many outliers off the blue shade in the subgraph of Figure 1. Could the authors give some interpretation of why there are so many outliers?"
>
> A: Although the negative relation has been verified in Figure 1, there are still noises in the data and some randomness in the optimization process. Nevertheless, as Figure 1 reports (see the upper right of each subfigure), the correlation coefficient shows that adversarial transferability and interaction have a strong negative relation.
> - - -
> Q5: **Optimizing interaction contradicts idea of one-pixel attack.** "For Equation 4, the value of the expected interactions is equivalent to the expectation of the contribution for each pixel. The authors aim to minimize the value of the expected interactions that is the same to average the contribution to all pixels. That may contradict the idea of the one-pixel attack. More interpretation should be given to understand the concept of interactions."
>
> A: Good questions.
> * **Responses to whether the expectation of interactions is equivalent to the expectation of the contribution:** Sorry for the confusion, but according to Eq. (13), the expectation of interactions is **NOT** equivalent to the expectation of the contribution. As Eq.(13) shows, the interaction between perturbation units $i, j$ can be written as the difference of contributions of $i$-th unit's contribution $\phi_i$,  when the $j$-th unit is perturbed *w.r.t* the case when the $j$-th unit is not perturbed. According to *the efficiency axiom* of the Shapley value (please see Appendix A for details), the expected contribution to all pixels is $\frac{1}{n}(v(\Omega)-v(\emptyset)$), which is not equal to the value of the expected interaction in the Eq.(4).
>
> * **Responses to whether our research conflicts with the one-pixel attack:** Our method does not conflict with one-pixel attack. More specifically, the contribution of each pixel $\phi(i \mid \Omega)$ can be decomposed as
>
> $$
> \phi(i \mid \Omega)=v(\{i\})-v(\emptyset)+\left[E_{j \in \Omega \backslash\{i\}}\sum_{s=0}^{n-2}\left[\frac{n-1-s}{n} E_{|S|=s, S \subseteq \Omega \backslash\{i, j\}}[v(S \cup\{i, j\})-v(S \cup\{i\})-v(S \cup\{j\})+v(S)]\right]\right]
> $$
>
> This does not conflict with the one-pixel attack. It is because the overall contribution of each pixel consists of two terms, (1) the pixel-wise attribution without interactions, i.e., $v(\{i\})-v(\emptyset)$, and (2) pixel-wise interactions, i.e. $\left[E_{j \in \Omega \backslash\{i\}}\sum_{s=0}^{n-2}\left[\frac{n-1-s}{n} E_{|S|=s, S \subseteq \Omega \backslash\{i, j\}}[v(S \cup\{i, j\})-v(S \cup\{i\})-v(S \cup\{j\})+v(S)]\right]\right]$. To this end, the one pixel-attack is mainly caused by the pixel-wise attributions without interactions.
>
>
> * **Responses to "more interpretation":** The psychical meaning of interactions can be understood as the collaborative behaviors of perturbation units that make themselves important in the attack. Please see the last paragraph on Page 3 for details.

---

> ### Author Response · Authors · 2020-11-19
> **Responses to Reviewer #4 (Part 2)**
>
> Q6: **Justify more transferability-boosting methods reduce interactions:** "The authors claim to provide a unified view to understand the enhancement of transferability; however the authors only explain three baseline models. Other types of adversarial example generation methods are not considered, such as Translation-Invariant Attack that the authors already mentioned in the related work. What is more, the Translation-Invariant Attack aims to compose the gradients of neighboring pixels together, which is contradictory to the authors’ idea of reducing the interaction of perturbations."
>
> A: We have conducted **new experiments** to empirically demonstrate that DI and TI, which enhance the adversarial transferability, actually reduce interactions. Please see Appendix O Table 12 for details.
>
> Besides, for the theoretical proof, we have theoretically analyzed the MI, VRA, SGM. However, for other methods of improving adversarial transferability, such as Diversity Input (Xie, et al., 2019), which uses random data augmentation during attacking, it is difficult to mathematically prove that they essentially reduce interactions. As for Translation-Invariant Attack, it composes the gradients of neighboring pixels together, which can be regarded as to smooth the gradient. Intuitively, the effect is similar to Variance-Reduced Attack, which decreases the interaction.
>
> ----
> Q7: "From the experiment results in Table 1, why the performance of the IR attack is worse than the baseline when attacking RN-152 and DN-201 in the ensemble setting?"
>
> A: A good question. It is mainly due to the model similarity between the source DNN and the target DNN. Particularly, our ensemble model is RN-34+RN-152+DN-121, so when the target model is RN-152, it is actually a white-box setting, instead of being a black-box. Therefore, the IR attack may be worse than the baseline.
>
> Moreover, we have conducted **new experiments** using three other target DNNs with different architectures from RNs or DNs, and updated the results to Table 2. Please see Table 2 for details. As Table 2 shows, perturbations generated on the ensemble model by IR attack were actually more transferable.
> - - -
> Q8: **About experimental settings.** "In Section 5, Experiments, baselines: "the transferability of each baseline was computed based on the best adversarial perturbation during the 100 steps…” and you also mention that "Previous studies usually set the number of steps to 10 or 20”. I am confused about why you set the step to be 100? For a fair comparison, you should follow the setting in previous studies."
>
> A: A good question. The fairness of comparisons is just the right motivation for us to evaluate the transferability using a leave-one-out strategy. To this end, our experimental settings ensure more fairness of comparisons than previous studies.
>
> First, the unfairness of using a fixed step number (such as 10 or 20 steps) for comparison in previous studies has been illustrated in Figure 7. As Figure 7 in Appendix K shows, the best transferability may be achieved in an intermediate step, rather than in the last step. If we set the number of steps to 20, it would be unfair for the MI attack, which achieves the best transferability in about the 10-th step. Please see Appendix K for details. This is a typical problem in adversarial examples. Previous studies[1] also discuss a similar issue that happened in adversarial training.
>
> Therefore, to enable fair comparisons, we do not directly use perturbations after 100 steps for evaluation. Instead, we use the leave-one-out evaluation, which automatically searches the best step in 100 steps.
>
> In addition, setting a fixed step number also has other problems in implementation. It is because previous studies do not have unified settings of hyper-parameters, which significantly boosts the difficulty for fair comparisons. For example, in the MI attack, the number of steps is set to 10 (with step size set to 1.6/255), while in the VR attack, the number of steps is 5 (with step size set to 4/255). Therefore, it is unfair to directly compare the MI attack with the VR attack, when we require them to optimize the perturbation for 10 steps.
>
> [1] Rice, Leslie, Eric Wong, and J. Zico Kolter. "Overfitting in adversarially robust deep learning." arXiv preprint arXiv:2002.11569 (2020).

---

> ### Author Response · Authors · 2020-11-19
> **Responses to Reviewer #4 (Part 3)**
>
> Q9: **About the range of $\lambda$.** "But in Figure 3, part a, the range of $\lambda$ is from 0 to 1.2 for DN121, which does not include the $\lambda$ value the authors suggested."
>
> A: We have followed your suggestions to conduct **new experiments** to test the effects of the larger $\lambda$ value. Experimental results show that the large $\lambda$ value usually exhibits high transferability. Please see Appendix P for details.
> - - -
> Q10: **About the time cost.** "The computational cost is not discussed. As the authors said in Section 5 "the computational cost of the IR loss is intolerable”. No discussion about the running time is provided. Moreover, the authors choose the step to be 100, which further increases the running time."
>
> A: The complexity of the expectation of the interaction is actually linear. Moreover, we have followed your suggestions to conduct **new experiments** to analyze the running time of the IR attack. Please see Appendix J.3 for more details. On the setting of the step number, we have a detailed discussion in the answer to Q8.
> - - -
> Q11: "The third line on page 5, the term $|\delta|^p_p$ is wrong."
>
> A: A good question. In fact, this is a controversial issue in the adversarial attack. A typical case is the C&W attack (Carlini & Wagner, 2017). The C&W attack defined the attack in the thirteenth equation in the paper using the norm constraint as $|\delta|_p$. However, the C&W method used the term of $|\delta|^p_p$ when it implemented the attack. Please see Section VI.A in (Carlini & Wagner, 2017) for this. Therefore, we believe both $|\delta|^p_p$ and $|\delta|_p$ are reasonable regularization for the attack.

---

### Official Review · AnonReviewer2 · 2020-10-28

**Rating:** 5
**Confidence:** 3

**Review:**

The paper mainly deals with the negative correlation between adversarial transferability and the interaction inside adversarial perturbations. The authors claimed that utilizing the correlation can be regarded as a unified perspective to understand previously proposed methods. To this end, they presented an adversarial attacking loss, which can directly reduce the interaction, defined as the individual interaction between two perturbation units to the total reward function.

This paper provides an intriguing perspective that can explain adversarial attacking mechanisms. Although I am not sure about the exact settings that draw the numbers, the empirical results seem good; thus, reducing the interaction inside perturbations could be useful in practice. However, I cast doubt on the theoretical significance of the central hypothesis. This concern derives from lacking analyses on 1) a valid (theoretical) reason for the negative correlation, 2) qualitative benefits of directly reducing the interaction based on the game theory.

[Quality]

The paper follows one solid logical structure aforementioned in Abstract and Introduction. However, I think Section 3 & 4 are confusing, which indeed are the core parts. For example, the propositions are overly verbose and hard to follow. Also, the theoretical statements and interpretations are not well aligned. I defer the detailed comments below.

[Originality]

The originality of the paper is not outstanding, but sufficient for acceptance.

[Significance]

The paper reveals one aspect of adversarial transferability for sure. However, bringing the game theory-based approach in this domain is quite unconventional. Hence, I strongly feel there should've been more theoretical arguments that strengthen the significance of this approach. I guess the authors' intention was to prove their claims empirically, and the paper provides some good results. Unfortunately, I feel this work is reluctant to compare the IR method with the State-of-the-Art methods (such as "MI vs. vanilla IR," "VR vs. vanilla IR," "SGM vs. vanilla IR").


[Comments & Questions (sorted by priority)]
1. As far as I understand from Proposition 1, it says that "multi-step attacks generate more interaction than single-step attacks." Please elaborate that this statement can be generalized to the following: "the adversarial transferability and the interactions inside adversarial perturbations are negatively correlated."
2. In Table 3, why is the result of (vanilla) IR Attack not reported? The table shows that HyridIR, a combination of all techniques, achieves the best performance. Hence, it implies that reducing the interaction in multiple ways can be stacked for achieving good performance. However, the table does not show the success rate of the pure IR attacking method. My concern is whether solely applying the IR attacking method has a clear contribution to be accepted in this conference. Please describe how IR attack (using only Eqn. 5) will perform compared to the SGM attacking method, which seems to be one of the State-of-the-Art.
3. According to the definition (Eqn. 3), the actual computation of the interaction is not very scalable when the set of players (\Omega) is large. I think Eqn. 4 also lacks scalability because it is natural to think that the set of the adversarial perturbations is a continuous space.
4. Is high $\lambda$ always effective? In Fig. 3, how the success rate changes when $\lambda = 10^2, 10^3, \dots$? In other words, I want to know when the tendency of Fig. 3 (a) converges to Fig. 3 (b) as $\lambda$ goes higher.
5. (Similar to #2) Why are the results of MI, VR, SGM methods not presented in Table 1?
6. Some of the arguments, especially propositions, are unnecessarily lengthy and hard to follow. To improve readability, I suggest clarifying the main points in the theoretical arguments. For example, please remove the redundant part "Given an input image $x \in\mathbb{R}^n$ ... $\delta^m_{multi}), y)$." in Props. 1, 2 & 3 and make this statement as a proper Definition.

---

> ### Author Response · Authors · 2020-11-19
> **Responses to Reviewer #2 (`Part 1)**
>
> Thank you very much for your careful review and constructive comments. We try our best to answer all your concerns.
>
> Q1: **About experimental settings.** "Although I am not sure about the exact settings that draw the numbers, the empirical results seem good."
>
> A: Experimental settings corresponding to results in Table 1-4 are introduced in the *Experiments* paragraph in Section 5. Let us take the $L_\infty$ attack in Table 1 for example. We conduct 100-step untargeted attacks under $\epsilon=16/255$ $L_\infty$ norm constraint using 1000 images in the ImageNet dataset on a pretrained source DNN to generate adversarial examples. Then, we use the leave-one-out evaluation to measure the success rates of 1000 images generated on the source DNN.
>
> Experimental settings corresponding to results in Figure 1 are introduced on the last paragraph on Page 4 and the second paragraph in Appendix G. We gradually change the value of of $c$ in $\min_\delta -\ell(h(x+\delta), y) + c\cdot \|\delta\|^p_p \; \text{s.t.} \; x+\delta\in[0, 1]^n$ to generate different adversarial examples on a source DNN. We require all adversarial attacks to generate perturbations with the same $L_2$ norm as the first adversarial attack. This is the stopping criteria for these adversarial attacks to ensure fair comparisons. Then, we measure the average interaction and the average transfer utility (on a target DNN) and draw each subfigure and report the Pearson correlation between the interaction and the transfer utility.
>
> Q2.1: "However, bringing the game theory-based approach in this domain is quite unconventional. Hence, I strongly feel there should've been more theoretical arguments that strengthen the significance of this approach."
>
> Q2.2: "Lack of a valid (theoretical) reason for the negative correlation."
>
> A: First, the theoretical proof serves as an explanation and verification for the phenomena of the negative correlations observed in experiments. Although the proofs of propositions are based on certain assumptions, these propositions still support the negative correlation between the interaction and the adversarial transferability.
>
> Second, Proposition 1 just serves as a strong motivation for us to propose the hypothesis of the negative correlation. Proposition 1 proves that the multi-step attack generates perturbations with larger interactions than the single-step attack. Xie et al. (2019) find that the multi-step attack tended to overfit and generate perturbations with lower transferability. Intuitively, large interactions mean a strong cooperative relationship between perturbation units, which indicates the significant over-fitting to the source DNN. Thus, this motivates us to propose the *hypothesis* that "the adversarial transferability and the interactions inside adversarial perturbations are negatively correlated." Please see the second paragraph under Proposition 1 on page 4 for details.
>
> Third, we have proved that three state-of-the-art transferability-boosting attacks essentially decrease interactions. This further supports the negative correlation observed in experiments.
>
> Fourth, we have verified the negative correlation from the other two perspectives, (1) we have conducted experiments to compute the transfer utility and the interaction of adversarial perturbations over different DNNs. Figure 1 shows the negative correlation between the transfer utility and the interaction; (2) Experiments have shown that the proposed IR loss can boost the transferability of PGD, MI, SGM, VR, and TI attacks. Please see Tables 1-4 and Tables 7-9 for details.

---

> ### Author Response · Authors · 2020-11-19
> **Response to Reviewer #2 (Part 2)**
>
> Q3.1: "Unfortunately, I feel this work is reluctant to compare the IR method with the State-of-the-Art methods."
>
> Q3.2: "Lack of qualitative benefits of directly reducing the interaction based on the game theory."
>
> Q3.3: "In Table 3, why is the result of (vanilla) IR Attack not reported? My concern is whether solely applying the IR attacking method has a clear contribution to be accepted in this conference. Please describe how IR attack (using only Eqn. 5) will perform compared to the SGM attacking method, which seems to be one of the State-of-the-Art."
>
> Q3.4: "Why are the results of MI, VR, SGM methods not presented in Table 1?"
>
> Answers to Q3.1-Q3.4: The above questions concern the comparisons between the IR loss and other methods.
>
> First, we have followed your suggestions to conduct **new experiments**, in order to explore the qualitative benefits of using the IR loss compared with state-of-the-art methods. In new experiments, we use the IR loss to boost the performance of the four baselines, including the MI, VR, TI, and SGM. Experimental results show that the IR loss successfully boosts the performance. Please see Table 3 in Section 5 and Table 7-9 in Appendix N.1 for details.
>
> Second, another issue about above experiments is whether the IR loss itself is powerful enough to compete with other methods (which also decrease interactions). Crucially, we have proved that all baselines, including the MI, VR, SGM attacks decrease the interaction in Section 4. Therefore, the comparison with the MI, VR, and SGM methods can just prove the optimization efficiency of decreasing the interaction, instead of examining whether the interaction reduction can boost the adversarial transferability.
>
> **To this end, although our proposed IR loss has achieved superior performance, strictly speaking, a good metric is not necessarily equivalent to a good loss of boosting adversarial transferability.** For example, the IOU metric is a standard metric to evaluate the detection accuracy, but it usually is not used as a loss function. Besides a good metric, the computation of transferable perturbations should also overcome the optimization problem (e.g., the local minimum of the interaction-decreasing task), which presents significant challenges to state-of-the-art methods. To this end, previous studies such as MI, VRA, SGM use various ways to optimize the interaction-decreasing problem. Therefore, we use the proposed IR loss to further boost the performance of existing transferability-boosting methods, which has successfully verified the negative correlation between adversarial transferability and interactions. Please see Tables 1-4 and Tables 7-9 for details.
>
> We can understand the behaviors of the proposed IR loss as follows. Different methods generate adversarial perturbations in different manifolds, thereby exhibiting different transferability. Based on the current perturbation, the IR loss can point out the optimization direction towards the further decrease of interactions in a local manner due to its optimization power. Thus, the IR loss further boosts the transferability.
>
> We have added the discussion about the behaviors of the proposed IR loss on the last paragraph of Page 8.
> - - -
> Q4: **About the relationship between Proposition 1 and the negative correlation.** "As far as I understand from Proposition 1, it says that 'multi-step attacks generate more interaction than single-step attacks.' Please elaborate that this statement can be generalized to the following: 'the adversarial transferability and the interactions inside adversarial perturbations are negatively correlated'."
>
> A: A good question. Both Proposition 1 and the finding in (Xie et al. 2019) are the backgrounds of the hypothesis that "the adversarial transferability and the interactions inside adversarial perturbations are negatively correlated." Specifically, Xie et al. (2019) find that multi-step attacks tend to exhibit lower transferability than single-step attacks. Moreover, Proposition 1 shows that multi-step attacks generate more interactions than single-step attacks. These two statements are the motivation for us to propose the hypothesis that "the adversarial transferability and the interactions inside adversarial perturbations are negatively correlated." Please see the second paragraph under Proposition 1 on page 4 for details. This hypothesis has been verified from three perspectives.
> * We have conducted experiments to compute the transfer utility and the interaction of adversarial perturbations over different DNNs. Figure 1 shows the negative correlation between the transfer utility and the interaction.
> * We have theoretically proved that three classic transferability-boosting methods essentially decrease interactions between perturbation units. Please see Section 4 for details.
> * Experiments have shown that the proposed IR loss can boost the transferability of the PGD, MI, SGM, VR, and TI attacks. Please see Tables 1-4 and Tables 7-9 for details.

---

> ### Author Response · Authors · 2020-11-19
> **Responses to Reviewer #2 (Part 3)**
>
> Q5: **About the scalability.** "According to the definition (Eqn. 3), the actual computation of the interaction is not very scalable when the set of players (\Omega) is large. I think Eqn. 4 also lacks scalability because it is natural to think that the set of the adversarial perturbations is a continuous space."
>
> A: The review concerns two kinds of scalability of the interaction.
>
> First, **is the computational cost of the proposed IR loss unaffordable when the number of players is large?** To this end, we have proved in Eq. (4) that the computational complexity of the expectation of the interaction is linear, which is scalable. In fact, we do not directly compute interaction using Eq. (3). Instead, we compute the expectation of interactions with Eq. (4).
>
> In addition, we have conducted a **new experiment**, in which the 100-step IR Attack on an ImageNet image is less than 50 seconds, which also prove the scalability of the interaction. Please see Appendix J.3 for details.
>
> Second, **is the computational cost affordable when we consider the continuous space of adversarial perturbations?** It has been widely discussed (Ancona et al., 2019; Sundararajan & Najmi,2019) that when applying the Shapley value, the feature space is regarded as binary. It is because (Sundararajan & Najmi, 2019) has shown that only in the binary space, the Shapley value is the unique method to satisfy the *linearity axiom, the dummy axiom, the symmetry axiom, and the efficiency axiom*.  Thus, when we compute the interaction, the perturbation can be regarded in the binary space, i.e., whether the perturbation unit is added to the input or not, which enables scalability.
> - - -
> Q6: **About different settings of $\lambda$.** "In other words, I want to know when the tendency of Fig. 3 (a) converges to Fig. 3 (b) as $\lambda$ goes higher."
>
> A: We have followed your suggestions to conduct **new experiments** to test the effects of the larger $\lambda$ value. Experimental results show that the large $\lambda$ value usually exhibits high transferability. Please see Appendix P for details.
>
> - - -
> Q7: **About readability.** "Some of the arguments, especially propositions, are unnecessarily lengthy and hard to follow. To improve readability, I suggest clarifying the main points in the theoretical arguments. For example, please remove the redundant part 'Given an input image...' in Props. 1, 2 & 3 and make this statement as a proper Definition."
>
> A: We have followed your suggestion to polish the language of the propositions.

---

### Official Review · AnonReviewer1 · 2020-10-28
**Interesting work and solid analysis**

**Rating:** 10
**Confidence:** 4

**Review:**

This work proposes that the transferability of adversarial attacks has a negative correlation with the interaction within an input perturbation. By defining the interaction of perturbations with the Sharpley value, it can quantify the interactions and demonstrate the negative correlation with the transferability. Furthermore, this work shows that prior work on adversarial attacks (e.g., VR attack and MI attack) can be explained by the (expected) interaction scores. This work further demonstrates that the way of enhancing transferability by minimizing the interaction within input perturbations, with the experiments on the image classification task.

Overall, I think this work provides a new perspective of understanding transferability and presents solid analysis/experiments to verify the hypothesis.

Only one comment about the definition of interaction scores. In some literature [Lundberg et al., 2019], it is called the Shapley interaction index, which uses the definition in equation (13) of Appendix D. Shapley interaction index has mainly been used in the machine learning literature recently for explaining feature interactions within models. E.g.,

1. Lundberg et al. Consistent Individualized Feature Attribution for Tree Ensembles. 2019
2. Chen and Ji. Learning Variational Word Masks to Improve the Interpretability of Neural Text Classifiers. 2020

---

> ### Author Response · Authors · 2020-11-19
> **Responses to Reviewer #1**
>
> Thank you very much for your careful review and constructive comments. We try our best to answer all your concerns.
>
> Q1: **About the formal name of the interaction.** "Only one comment about the definition of interaction scores. In some literature [Lundberg et al., 2019], it is called the Shapley interaction index."
>
> A: Thank you for your comments. First, yes, the interaction used in our paper is the Shapley interaction index, and we have clearly cited the corresponding paper (Michel & Marc, 1999) in Section 3.1. Second, we have revised the paper and used the formal name "Shapley interaction index." Third, you have mentioned another paper that used the Shapley interaction index in the NLP task. We have added discussions about this paper in our related work to further point out the broad applicability of the Shapley interaction index.

---

### Official Review · AnonReviewer3 · 2020-10-29
**This work defines interaction of coordinates in the input data based on Shapley values, and shows a negative correlation between the adversarial transferability and the interaction.**

**Rating:** 6
**Confidence:** 3

**Review:**

The paper presents a negative correlation between the adversarial transferability and the interaction between coordinates.  The interaction is defined by the Shapley values used in the game theory to measure the contribution of players.  The author(s) defined the interaction between coordinates (players) as the difference between the joint contribution and the sum of conditional contributions.  They show a negative correlation between the adversarial transferability and the interaction with various known adversarial attacks.

This work also shows an improvement in the transferability of adversarial perturbations by incorporating the interactive loss with the classification loss.

What is missing in the paper is the reason and motivation for using the Shapley values for defining the interaction.  No comparison and discussion is given the difference between their method and other methods that define the interaction between variables.

---

> ### Author Response · Authors · 2020-11-19
> **Response to Reviewer #3**
>
> Thank you very much for your careful review and constructive comments. We try our best to answer all your concerns.
>
> Q1: **About the motivation.** "What is missing in the paper is the reason and motivation for using the Shapley values for defining the interaction."
>
> A: A good question. We have added reasons and motivations in Appendix A.2.
> 1. **The interaction based on the Shapley value is theoretically rigorous.** We use the interaction defined based on the Shapley value, because the Shapley value has a solid theoretical foundation in the game theory, which is the *unique*  attribution satisfying four desirable axioms, i.e., *the linearity axiom, the dummy axiom, the symmetry axiom*, and *the efficiency axiom*.
>
> 2. **The metric defined based on the Shapley value does not depend on network architectures.** Because adversarial transferability is a general property for the attack, a convincing metric for adversarial transferability is supposed not to be directly related to the network architecture. To this end, the computation of the interaction based on the Shapley value does not depend on the network architecture.
>
> 3. **The computational cost of the Shapley-based interaction-reduction loss is relatively low.** Because of *the efficiency axiom* of the Shapley value, we prove that the time cost of computing the IR loss $\ell_{interaction}=-\frac{1}{n-1} \mathbb{E}_{i}[v(\Omega)-v(\Omega \backslash\{i\})-v(\{i\})+v(\emptyset)]$ is linear, i.e., $O(n)$, where $n$ is the dimension of features. The linear complexity makes it possible to apply the interaction to high-dimensional data and deep neural networks.
> - - -
> Q2: "No comparison and discussion is given the difference between their method and other methods that define the interaction between variables."
>
> A: A good question. We have added discussions about the difference between our method and other methods that define the interaction in Appendix A.2.
>
> Different metrics measure the interaction from different perspectives and represent different meanings. The interaction used in our paper measures the interaction between the units $i,j$ as the change of the $i$-th unit’s importance when the unit $j$ exists *w.r.t* the case when the unit $j$ is absent. In comparison, Daria Sorokina (2008) defined the interaction of $K$ input variables of additive models. Tsang et al. (2018) measured statistical interactions based on DNN weights. Jin et al. (2020) quantified the contextual independence of words to hierarchically explain the LSTMs. This has been clarified in the related work section.
>
> In addition, whether the interaction metric is computed based on the network architecture is a key difference between our metric and other interaction metrics. The computation of the interaction used in this paper does not depend on the network architecture. In comparison, previous definitions of the interaction are usually oriented to some specific model architectures. For example, the interaction proposed by Tsang et al. (2018) requires the DNN to be fully-connected. The interactions proposed by Murdoch et al. (2018) and Jin et al. (2020) are designed for LSTMs. The Hessian-based interaction (Janizek et al., 2020) requires the DNN to use the SoftPlus operation to replace the ReLU operation.
>
> Furthermore, the interaction used in our paper is defined based on the Shapley value, which is the *unique* attribution satisfying four desirable axioms, i.e., *the linearity axiom, the dummy axiom, the symmetry axiom*, and *the efficiency axiom*.

---

### Author Response · Authors · 2020-11-19
**Summary of Changes**

We would like to thank all the reviewers for their careful reviews and valuable comments. Based on reviewers' feedback, we have updated the paper, with the following revisions:
1. We have conducted a new experiment, in which we used different random samplings of grids or different initial perturbations to compute the variance. Please see Tables 1-4 for details.
2. We have conducted a new experiment using the MI+IR attack, in order to further demonstrate the effectiveness of the IR loss. Experimental results have shown that the IR loss further enhanced the performance of the MI attack. Please see Table 7 and Appendix N.1 for details.
3. We have conducted a new experiment using the VR+IR attack, in order to further demonstrate the effectiveness of the IR loss. Experimental results have shown that the IR loss further enhanced the performance of the VR attack. Please see Table 9 and Appendix N.1 for details.
4. We have conducted a new experiment, in which we conducted targeted attacks on the CIFAR-10 dataset. Please see Table 10 and Appendix N.3 for details.
5. We have conducted a new experiment on the ensemble source model, in which we added three additional DNNs as target DNNs. Please see Table 2 and the "Experiments" paragraph in Section 5 for details.
6. We have conducted a new experiment about the computational cost. We measured the time cost of generating perturbations using the IR Attack in Table 6. It shows that the IR Attack was computationally applicable to high-dimensional data and deep neural networks. Please see Table 6 and Appendix J.3 for details.
7. We have conducted a new experiment, in which we measured the interaction inside adversarial perturbations that were generated by the DI attack and the TI attack, respectively. We demonstrated that the DI and the TI attack both reduced interactions. Please see Table 11 and Appendix O for details.
8. We have conducted a new experiment in order to test the effects of different $\lambda$ values. Please see Appendix P for details.
9. We have revised the paper to use the formal name "Shapley interaction index" to term the interaction used in our paper.
10. We have added more discussions about related works on interactions. Please see the "Interaction" paragraph of Section 2 for details.
11. We have added a paragraph to explain the enhancement of the adversarial transferability caused by the proposed IR loss. Please see the last paragraph on Page 8.
12. We have revised Tables 1-2 by replacing the term "Baseline" of $L_\infty$ attacks and $L_2$ attacks by "PGD $L_\infty$" and "PGD $L_2$" for clarification, respectively. Similarly, we have renamed "IR Attack" of $L_\infty$ attacks and $L_2$ attacks by "PGD $L_\infty$+IR" and "PGD $L_2$+IR" for clarification, respectively.
13. We have added a subsection "Motivation" within Appendix A to clarify the motivations of using the Shapley interaction index to define the interaction. Please see Appendix A.2 for details.
14. We have added some paragraphs in Appendix N.1 to introduce the implementation details of the VR+IR Attack. Please see the second and third paragraphs in Appendix N.1 for details.
15. We have fixed the reference format problem of Figure 4 and Figure 7 in the Appendix.
16. We have reorganized the Appendix to improve the readability. We have merged the previous sections Appendix G and Appendix I into Appendix J in this version for better presentation.
17. We have added a subsection "scalability of the interaction loss" within Appendix J to discuss two kinds of scalability of the proposed IR loss. Please see Appendix J.3 for details.
18. We have polished the language according to suggestions from the reviewer. We have also cited and discussed several new related studies, which were mentioned by reviewers, in the related work section.

---

### Decision · Program_Chairs · 2021-01-07
**Final Decision**

**Decision:**

Accept (Poster)

**Comment:**

This work provides interesting insights on the transferability of adversarial perturbations and proposes ways of making it more effective. While several reviewers have found parts of the paper unsatisfactory, there are interesting results to merit acceptance.